# HIV vaccine candidate ΔV1gp120 formulated in ALFQA adjuvant augments mucosal immunity in female macaques

Simian or Human immunodeficiency virus (SIV or HIV) vaccines based on V1-deleted envelope virus-like particles, delivered by the DNA/ALVAC platforms, followed by the ΔV1gp120 boost formulated in Alum, protect 50% and 80% of macaques from mucosal infection with $SIV_{mac251}$ or Simian-Human immuno-deficiency virus, respectively. Adding the Army Liposome Formulation + QS21 (ALFQ) adjuvant to the ΔV1gp120+Alum boost (ALFQA) may enhance protective immune responses. Here, we show that ALFQA protects 58% of female macaques from infection following eleven exposures to $SIV_{mac251}$, achieving 79% vaccine efficacy. The ALFQA vaccine regimen augments mucosal CD73+CD163+ M2-like macrophages and NKp44+ innate lymphoid cells (ILCs), while reducing NKG2A-NKP44- cells producing interferon-γ. Antibody-Dependent Cellular Cytotoxicity (ADCC) targeting helical V2, and mucosal tolerogenic dendritic cells-10 (DC-10) and envelope-specific interleukin-17+ NKp44+ ILCs, correlate with decreased risk of infection. Plasma proteome analysis links vaccine efficacy to lymphotoxin-α, mucosal DC-10, and chemokine (C-C motif) ligand-8, a chemokine produced mainly by M2-macrophages. These data support the role of pro-resolution immunity in protection afforded by the V1-deleted SIV and HIV immunogens. The Combined Long-term Efferocytosis and ADCC Responses (CLEAR) phase I HIV-vaccine trial is designed to test the safety and immunogenicity of the Alum and ALFQA adjuvants in combination with V1-deleted HIV immunogens in humans.

Vaccination is one of the most effective tools to prevent infectious diseases by eliciting immunity either to prevent infection by pathogens or by inhibiting the onset of disease[1]. The most common types of vaccines are: inactivated, live-attenuated, nucleic acid (DNA and mRNA), subunit, toxoid, viral vectors, and viral virus-like particles (VLPs) vaccines[1,2]. Subunit vaccines containing one or more antigens, administered as a recombinant protein or peptide, are safe, stable, and scalable[1]; however, they often require adjuvants to elicit protective immunity[3,4]. Adjuvants augment immunity by different mechanisms such as prolonging immunogen exposure (depot effect) or activating pathways on antigen-presenting cells (APCs), including Toll-like receptors (TLRs), the cyclic GMP-AMP synthase-stimulator of interferon genes (cGAS-STING), C-type lectin receptors (CLRs), and pattern recognition receptors (PRRs)[4,5]. Activation of these pathways leads to enhanced antigen presentation, cytokine and chemokine production, and ultimately to enhanced innate and adaptive immunity and T cell responses polarization[4,6–9]. Adjuvant substances approved by the United States Food and Drug Administration (FDA) routinely used in vaccines include synthetic small molecule compounds, particulate materials, or complex natural extracts[10]. The first adjuvant was discovered in 1926 by Alexander Glenny, a British immunologist who noticed the immune-enhancing effects of aluminum salts. The use of aluminum adjuvants in humans began in 1932, and they were the only approved adjuvants for seventy years[11]. Aluminum hydroxide

✉ e-mail: massimiliano.bissa@nih.gov; franchig@mail.nih.gov

[Al(OH)$_3$], commonly referred to as Alum, is the most widely used adjuvant in licensed vaccines. Alum enhances immune responses through different mechanisms[12], and it maintains the physical and chemical features of antigens (repository effect) while also managing protracted release of the immunogens, facilitating prolonged stimulation of the immune system. Moreover, Alum promotes the phagocytosis of antigens by antigen-presenting cells (APCs) and macrophages by modifying their activation status[13]. Alum also activates the nucleotide binding oligomerization domain (NOD) like receptor protein 3 (NLRP3) pathway, which leads to increased inflammation[14], mediates the differentiation of monocytes/macrophages into dendritic cells (DCs) that migrate to lymph nodes to induce T cell proliferation[13], and promotes type 2 T helper (Th2) CD4 T cell responses to target and destroy infected cells[15].

Adjuvants have become a pillar of immunization, and more recently, the FDA has approved additional types, such as oil-in-water emulsion adjuvants (e.g., MF59 and AS03); TLR agonist molecule-based adjuvants (AS04 and CPG ODN 1018); particulate adjuvants (AS01), and lipid nanoparticle (LNP) adjuvants[10]. In the 1980s, the Walter Reed Army Institute of Research (WRAIR) developed the army liposome formulation (ALF), constituted of liposomes containing saturated phospholipids, cholesterol, and monophosphoryl lipid A (MPLA)[16,17]. Subsequent modifications to ALF resulted in the development of ALFQ, obtained by the combination of high cholesterol-ALF and QS21 saponin. Adsorption of the antigen to an aluminum salt was mixed with either ALF or ALFQ to generate ALFA and ALFQA, respectively[18]. MPLA by itself is highly toxic. However, when MPLA is incorporated in ALF adjuvants, the toxicity of the fatty acyl chains is lost as they are incorporated into the liposomal bilayer. Importantly, MPLA binds to the TLR4/myeloid differentiation factor 2 (TLR4/MD-2) receptor complex, which upregulates the transcription of pro-inflammatory cytokines and Type I IFN signaling and also induces NLRP3 inflammasome activation to promote secretion of interleukins (IL) −1β and −18[19]. QS21 saponin has a flexible fatty acyl chain and includes eight sugars, which cover the polar region of QS21[18]. Data suggest that these sugars can bind different types of lectin receptors, which are present on innate cells, such as dendritic cells[18,20]. Studies in mice have demonstrated that immunization with immunogens formulated with ALFQA adjuvants induces higher antibody responses than Alum, and ALFQA generated a more balanced Th1/Th2 immune response, in contrast with the predominantly Th2 response induced by Alum[21]. Therefore, we hypothesized that the ability of ALFQA to promote both adaptive and innate immune responses may advance the development of an effective vaccine to prevent human immunodeficiency virus (HIV) acquisition. While anti-HIV vaccines have proven notoriously difficult to develop, the inclusion of proper adjuvants may enhance the viability of previously tested and novel candidate vaccine platforms. To date, of the nine phase IIb/III clinical trials conducted[22], only one demonstrated significant, albeit modest, vaccine efficacy of 31.2%, the RV144 Thai trial (31.2%)[23]. In this phase III trial, human volunteers were vaccinated with recombinant canarypox-derived poxvirus vector (ALVAC) expressing HIV clade B/AE env and gag/pro, and bivalent gp120 -TM env HIV clade B/AE proteins formulated in aluminum hydroxide adjuvant[23].

Similar studies conducted using the simian immunodeficiency virus mac251 (SIV$_{mac251}$) non-human primate (NHP) model recapitulated the efficacy of this vaccine strategy using Alum adjuvant, but not MF59[24], identified similar correlates of protection and improved vaccine efficacy by deleting variable region 1 (V1) from the envelope immunogens to enhance antigen binding to the V2 site[25–29]. The resulting vaccine platform will be tested in human volunteers in the combined long-term efferocytosis and antibody-dependent cell-mediated cytotoxicity (ADCC) Responses [CLEAR] phase I HIV-vaccine trial in 2026.

In the present study, we tested the impact of ALFQ in combination with V1-deleted (ΔV1) gp120 formulated in alum (ALFQA) and found that boosting with ΔV1gp120 in ALFQA resulted in a 79% decreased risk of SIV$_{mac251}$ acquisition, compared to a 59.8% decreased risk in animals vaccinated with ΔV1gp120 formulated in Alum alone. ALFQA augmented antibody levels to V2 and CD14$^+$ cell-mediated efferocytosis. In addition, ALFQA-induced plasma lymphotoxin-α (LTA), which correlated with both decreased virus acquisition and with mucosal tolerogenic DC-10 frequency. In turn, mucosal CD163$^+$ CD73$^+$(M2-like) macrophages correlated with efferocytosis, an immune response contributing to an anti-inflammatory mucosal landscape unfavorable to HIV/SIV seeding. Finally, the finding that the plasma levels of chemokine (C-C motif) ligand 8 (CCL8), a chemokine that binds to CC chemokine receptor type 5 (CCR5), correlated with a decreased risk of SIV infection suggests that CCL8 may directly inhibit viral infection.

## Results

### ΔV1DNA/ALVAC/DV1gp120/ALFQA vaccine reduces the risk of SIV infection

In macaques, the ΔV1DNA/ALVAC vaccine boosted with ΔV1M766 SIV$_{mac251}$ gp120 protein formulated in Alum reduces the risk of virus acquisition from intrarectal and intravaginal SIV$_{mac251}$ exposure by ~60% and leaves ~50% of animals uninfected[27–29]. We vaccinated one group of macaques (Fig. 1a) with the same ΔV1 vaccine regimens used in prior studies and administered the protein boost adjuvanted in alum plus ALFQ (ALFQA; $n = 12$). We then compared its efficacy and the resulting immune responses to those obtained from previously published studies in which macaques were immunized with the protein boost adjuvanted in Alum only ($N = 30$). All 42 vaccinated macaques were co-immunized at weeks 0 and 4 with SIV p57 gag and SIV ΔV1M766 gp160 env DNA plasmids to generate virus-like particles (VLPs). Priming was followed by two boosts with recombinant ALVAC-SIV viral vectors co-expressing gag-pro and SIV$_{mac251}$ wild-type M766 gp120- transmembrane (TM) at week 8 and 12. The ΔV1M766 SIV$_{mac251}$ gp120 protein formulated in Alum or ALFQA were given to the respective groups at week 12 in the contralateral thigh. At week 17, both vaccinated groups and a naïve control group ($n = 37$) were exposed intravaginally to 11 consecutive low doses of SIV$_{mac251}$. A significantly decreased risk of viral acquisition was observed in the Alum group when compared to naïve controls ($p = 0.0021$; Fig. 1b). The ALFQA group also showed significantly decreased risk of acquisition compared to naïve controls ($p < 0.0001$; Fig. 1c). Although the study was not powered to compare the two vaccinated groups, the comparison of the viral acquisitions between ALFQA and Alum groups showed a trend towards further delayed viral acquisition ($p = 0.11$; Fig. 1d), but this was not significant. Vaccine efficacy was 79 and 60% in the ALFQA and Alum groups, respectively. At the end of the challenge, 58% (seven protected, five infected) of animals in the ALFQA group and 33% (10 protected, 20 infected) in the Alum group remained protected. Vaccinated animals that became infected showed only a transient decrease in plasma viral load (VL) compared to naïve animals at multiple timepoints (Fig. 1e and Supplementary Fig. 1a–g). Overall, these results demonstrate the addition of ALFQ to Alum is efficacious. A study properly powered by increasing animal numbers would be required to confirm these data.

### ALFQA focuses antibody response to V2

Next, we focused on comparing protective antibody responses identified in prior studies using Aum alone[30–32]. We found that despite the total anti-envelope antibody titers in blood being higher in Alum than the ALFQA group ($p < 0.0001$; Supplementary Fig. 2a), the targeted antibody response to V2 peptide 26 following vaccination was higher in ALFQA ($p = 0.016$; Fig. 2a and Supplementary Tables 1, 2). Since prior studies have demonstrated that ALFQ can increase the avidity of antibodies[33], we investigated the avidity of antibodies targeting the

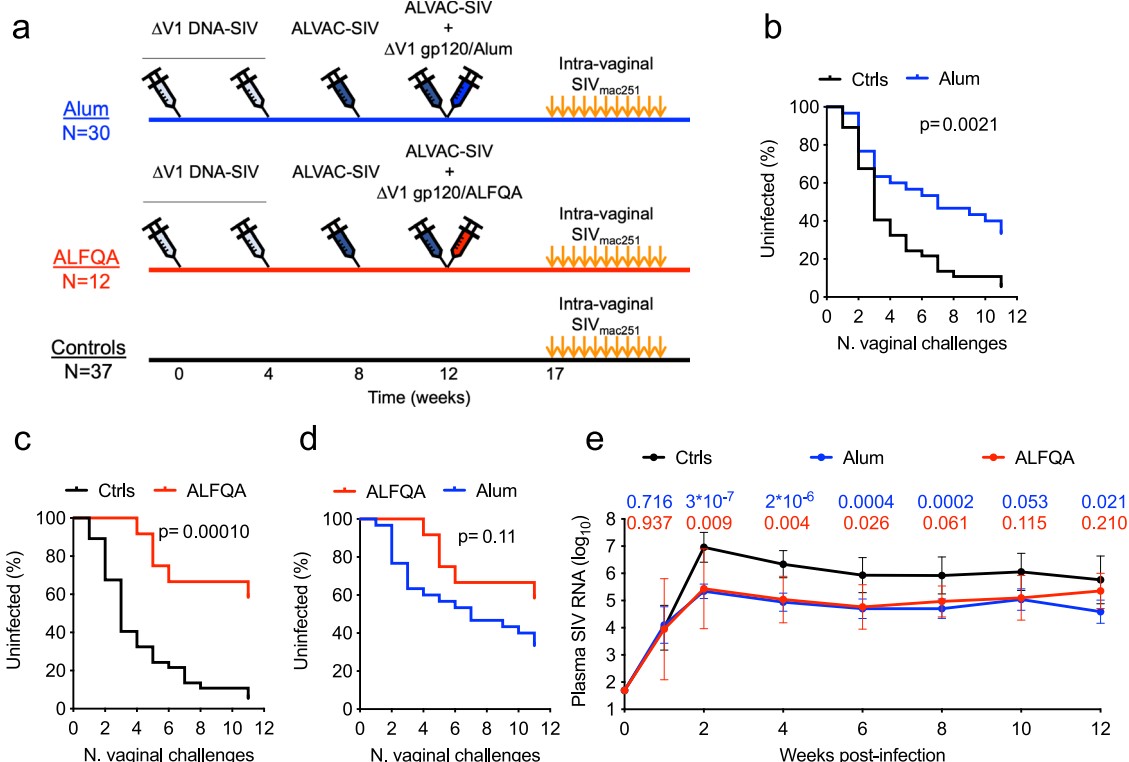

**Fig. 1 | Study design, vaccine efficacy, and viral load. a** Schematic study design of Alum (blue) and ALFQA (red) immunized groups with immunization schedule (weeks 0–12) and $SIV_{mac251}$ intravaginal challenges (weeks 17-27). **b–d** $SIV_{mac251}$ acquisition. The number of intravaginal exposures before infection was assessed in **b** Alum ($n = 30$) and **c** ALFQA ($n = 12$) animals relative to control ($n = 37$) animals, or **d** between Alum and ALFQA animals (Log-rank Mantel–Cox test). **e** $Log_{10}$ Simian immunodeficiency virus (SIV) RNA levels in plasma over time following $SIV_{mac251}$ infection (weeks; geometric mean with error and 95% CI) in Alum ($n = 20$), ALFQA ($n = 5$), and control ($n = 26$) animals. *P* values indicate a two-tailed Mann–Whitney *U* comparison test between the Alum (blue *p* values) or ALFQA (red *p* values) groups and the control group. Alum, ALFQA and control animals are depicted in blue, red, and black, respectively. Source data are provided as a Source Data file.

entire ΔV1gp120 or the cyclic V2 peptide (cV2, Biotin-Ttds-CIAQNNCTGLEQEQMISCKFNMTGLKRDKTKEYNETWYSTDNES-RCY-NH2) in the plasma of the ALFQA and Alum groups collected 2–3 weeks following the last immunization. While avidity scores of antibodies targeting ΔV1gp120 or cV2 peptide did not differ between alum and ALFQA groups (Supplementary Fig. 2b, c), the antibody response targeting V2 (peptide 26) strongly correlated with avidity score to cV2 in the context of ALFQA but not alum ($p < 0.001/ρ = 0.92$ and $p = 0.708/ρ = −0.15$, respectively; Fig. 2b). Neutralizing antibodies responses (inhibitory dilution 50, $ID_{50}$) against tier 1 A $SIV_{mac251}$ and $SIV_{smE660}$ measured at 2–3 weeks following last vaccination were similar in the two groups. In contrast, responses against tier 1B $SIV_{smE660}$ and the $SIV_{mac251}$ used for the challenge were lower in the ALFQA than in the alum group (Supplementary Table 3). In the ALFQA group, neutralizing antibodies did not correlate with risk of infection (Supplementary Table 3), whereas in the Alum group, neutralizing antibodies against the challenge virus negatively correlated with reduced risk of acquisition ($ID_{50}$; $p = 0.035/ρ = −0.62$). Interestingly, the ALFQA group exhibited a trend towards higher levels of total anti-envelope antibodies in vaginal mucosa secretions ($p = 0.082$; Fig. 2c).

Antibody-dependent cellular cytotoxicity (ADCC) mediated by antibodies targeting V2 is associated with decreased risk of SIV acquisition in the NHP model[27,29,34], so we investigated the effect of ALFQA on vaccine-elicited antibodies mediating ADCC in plasma samples collected following the last immunization. Plasma total and V2-specific ADCC did not differ (Supplementary Fig. 2d–f) between the ALFQA and alum groups, and in both groups, V2-specific ADCC correlated with a decreased risk of viral infection (ALFQA $p = 0.010/ρ = 0.73$; alum $p = 0.003/ρ = 0.53$; Fig. 2d), confirming the protective

role of ADCC in the ALFQA vaccination as well as alum. Interestingly, in the ALFQA group only, the plasma antibody titers to gp120 were associated with total ADCC killing (ALFQA $p = 0.020/ρ = 0.66$; alum $p = 0.516/ρ = −0.18$; Supplementary Fig. 2g), further confirming that ALFQA focuses humoral response to V2.

## ALFQA augments antibody-dependent phagocytosis and CD14[+] efferocytosis

Next, we investigated whether the protein boost adjuvanted with ALFQA could affect the ability of vaccine-elicited antibodies to mediate trogocytosis by analyzing plasma collected at baseline and at 2–3 weeks following vaccination. Trogocytosis is a mechanism of cell-to-cell interaction characterized by the exchange of membrane material, as one cell acquires fragments from another cell, that can be enhanced by antibodies and can remove viral antigens from the surface of infected cells, helping pathogens escape immune responses[35,36]. We observed higher levels of vaccine-induced trogocytosis targeting ΔV1gp120-coated cells in alum vaccinated macaques than the ALFQA group ($p = 0.014$; Fig. 2e). No such difference was found when the wild-type gp120 was used to coat cells (Supplementary Fig. 2h). In addition to ADCC, vaccine-elicited antibodies can mediate cellular phagocytosis. This Fc-mediated effector function is exhibited by both monocytes (antibody-dependent cellular phagocytosis; ADCP) and neutrophils (antibody-dependent neutrophil phagocytosis; ADNP), resulting in an antibody-dependent engulfment of infected cells[37]. We investigated the adjuvants' effects on ADCP and ADNP in plasma collected at baseline and at 2–3 weeks following the last vaccination. Indeed, ALFQA induced higher vaccine-induced ADCP than Alum, when plasma was tested on cells

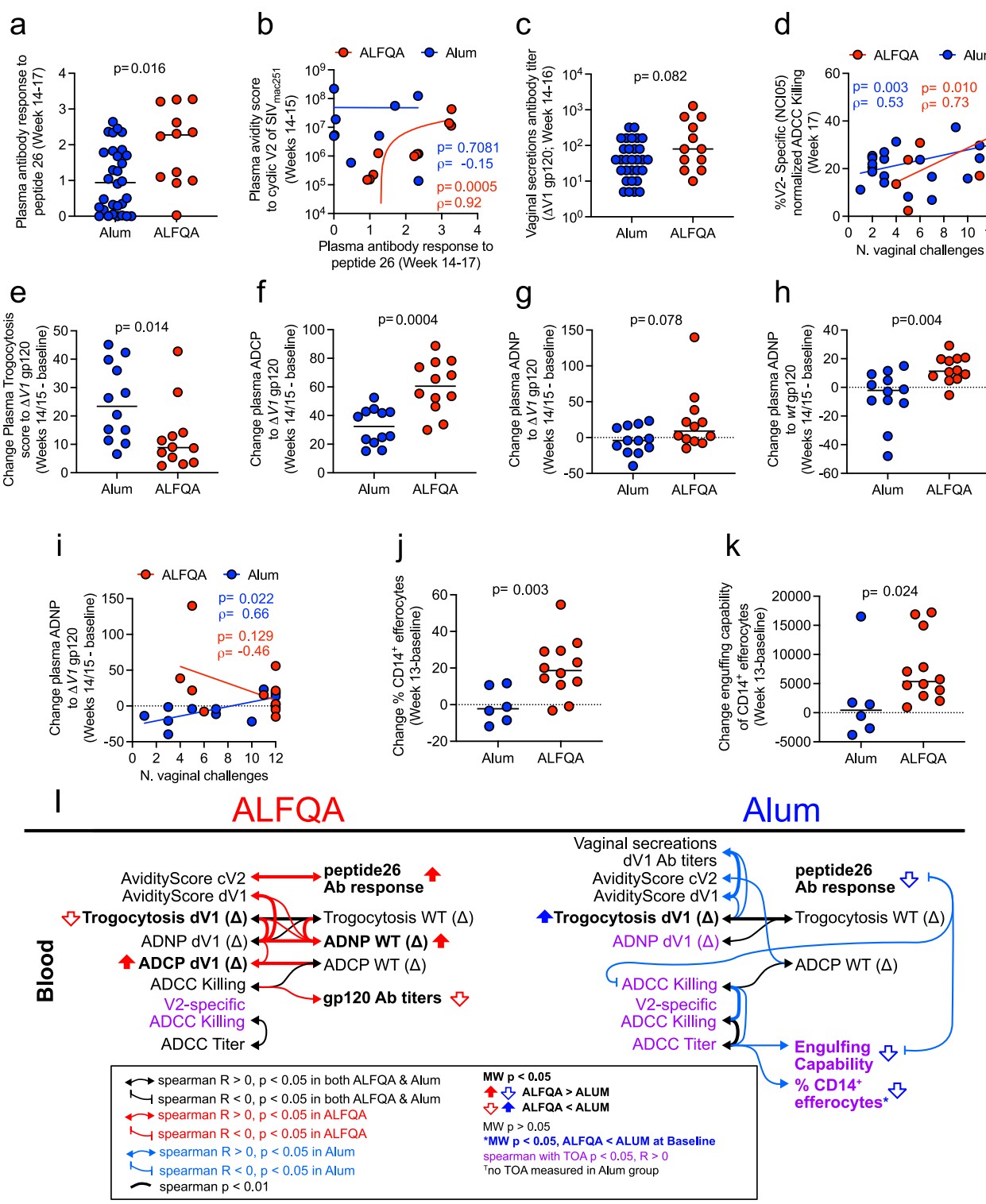

coated with ΔV1gp120 (*p* < 0.001; Fig. 2f) but were comparable when tested with the wild-type gp120 protein (Supplementary Fig. 2i). Phagocytosis exhibited by neutrophils trended higher when tested on ΔV1gp120-coated cells and was significantly increased on wild-type gp120-coated cells with ALFQA compared to Alum (*p* = 0.078 and *p* = 0.004, respectively; Fig. 2g, h). Interestingly, vaccine-induced ADNP targeting ΔV1gp120 significantly correlated with decreased risk of virus acquisition only in animals vaccinated with alum (alum *p* = 0.022/ρ = 0.66; ALFQA *p* = 0.129/ρ = −0.46; Fig. 2i).

In efferocytosis, monocytes and neutrophils mediate the clearance of apoptotic cells to dampen inflammation[38,39]. In prior studies,

we uncovered a role for vaccine-induced CD14⁺ cells mediated efferocytosis in vaccine efficacy that may complement V2-specific ADCC, a response that also induces apoptosis[28,29]. Therefore, we investigated the effect of ALFQA on the frequency of CD14⁺ (cluster of differentiation 14) cells mediating efferocytosis following vaccination. The analysis revealed that, in animals immunized with ALFQA, CD14⁺ cells isolated from blood at 1 week following the last vaccination had increased frequency of CD14⁺ cells mediating efferocytosis (*p* = 0.003; Fig. 2j) and a greater ability to engulf apoptotic neutrophils (*p* = 0.024; Fig. 2k), suggesting that ALFQA enhances the activity of blood CD14⁺ cells and promotes efferocytosis.

**Fig. 2 | Antibody responses, Fc receptor-dependent function of antibodies, and efferocytosis. a** Antibody response (optical densities, OD) to peptide 26 (variable region 2) in Alum ($n = 30$) and ALFQA ($n = 12$) animals at weeks 14–17. **b** Correlation between antibody response to peptide 26 and $Log_{10}$ plasma avidity score of antibodies to the cyclic V2 region of $SIV_{mac251}$ in alum ($n = 9$) and ALFQA ($n = 10$) animals at weeks 14–17. **c** Mucosal (vaginal secretions) antibody titers targeting the whole $\Delta V1gp120$ protein of $SIV_{m766}$ in Alum ($n = 30$) and ALFQA ($n = 12$) animals at weeks 14–16. **d** Correlations between V2-specific antibody-dependent cellular cytotoxicity (ADCC; NCI05 antibody) against gp120-coated cells and the time of acquisition (TOA) in alum ($n = 30$) and ALFQA ($n = 12$) animals at weeks 17. **e–g** Vaccine-induced change (week 14/15 – baseline) of **e** trogocytosis score, **f** antibody-dependent cellular phagocytosis (ADCP), and **g** antibody-dependent neutrophil phagocytosis (ADNP) against $SIV_{m766}$ $\Delta V1gp120$ protein measured in plasma of alum ($n = 12$) and ALFQA ($n = 12$) animals. **h** Vaccine-induced change (week 14/15 – baseline) of ADNP against $SIV_{m766}$ wild-type gp120 protein measured in plasma of alum ($n = 12$) and ALFQA ($n = 12$) animals. **i** Correlation between vaccine-induced change (week 14/15 – baseline) ADNP against $SIV_{m766}$ $\Delta V1gp120$ protein measured in plasma and Time of acquisition (TOA) in Alum ($n = 12$) and ALFQA ($n = 12$) animals. **j, k** Vaccine-induced changes (week 13 – baseline) of **j** the frequency of $CD14^+$ efferocytes and **k** their engulfing capability of apoptotic neutrophils in the blood of alum ($n = 6$) and ALFQA ($n = 12$) animals. **l** Schematic summarizing Spearman correlations among systemic antibody responses and functions in ALFQA animals (left) and Alum (right) animals. Associations of $p < 0.05$ are shown with a thin line, and $p < 0.01$ with a thicker line connecting variables. Associations found in both Alum and ALFQA are shown in black. Associations found only in ALFQA or Alum animals are in red or blue, respectively. Double-headed arrows indicate Spearman $R > 0$, vertical stubs indicate Spearman $R < 0$. Mann–Whitney $p < 0.05$ direction between groups is indicated by the vertical block arrows. Positive Spearman correlation with TOA $p < 0.05$ is depicted with purple text. Comparisons: **a, c, e–h, j, k** two-tailed Mann–Whitney $U$-test with median; Correlations: **b, d, i** two-tailed Spearman correlation with simple linear regression. Alum animals and correlations are depicted in blue, ALFQA animals and correlations are depicted in red. In panels **a–k**, SIV-immunized Alum and ALFQA animals are depicted as black circles filled in blue and red, respectively. Source data are provided as a Source Data file.

To assess the relationships among these responses in the context of ALFQA vs alum vaccination, we analyzed Spearman correlations among the vaccine-induced trogocytosis, ADNP, ADCP, ADCC, and efferocytosis metrics (Fig. 2l). Strikingly, vaccine-induced antibody-mediated ADNP, trogocytosis, and ADCP targeting $\Delta V1gp120$ were positively associated with each other in ALFQA animals only (Supplementary Fig. 2j, k), suggesting the ability of the two adjuvants to induce different Fc-mediated antibody functions. Further, in the ALFQA group, ADNP correlated with the avidity score of antibodies targeting $\Delta V1gp120$ (Supplementary Fig. 2l), suggesting that the higher level of antibodies mediating ADNP induced by ALFQA also have high avidity. Together, these analyses demonstrate that the addition of ALFQ to Alum changes the humoral immune response.

## ALFQA SIV $\Delta V1gp120$ boost increases mucosal $CD73^+CD163^+$ anti-inflammatory macrophages in macaques

Following virus exposure and infection, the clearance of apoptotic infected cells at the mucosal level is likely crucial to prevent further recruitment of pro-inflammatory cells that favor the spreading of the virus. We therefore investigated the presence of macrophages in rectal mucosal samples. Since M2-like polarized macrophages bear higher efferocytic capability[40], we first assessed the frequency of M2-like macrophages expressing the CD163 receptor in rectal mucosa by flow cytometry (Supplementary Fig. 3a) collected at baseline and 1 week following the last boost. CD163 is a surface marker associated with M2-like pro-resolving macrophages[41]. Because analysis of mucosal cells must be conducted on fresh samples, mucosal macrophages assessed in the ALFQA-treated group were compared to those measured in a simultaneous, ongoing NHP study using the same vaccine platform with Alum (Supplementary Fig. 4a). In both groups, mucosal samples collected at baseline and 1 week following the last immunization were analyzed with the same flow cytometry analysis. Although there was no difference in the vaccine-induced frequency of $CD163^+$ macrophages between the two groups (Supplementary Fig. 4b), the ALFQA group had a higher frequency of vaccine-induced $CD163^+$ macrophages expressing the ecto-enzyme CD73 than the Alum group ($p = 0.006$; Fig. 3a). Studies support the potential role of CD73 in mediating an efferocytosis-driven programming of macrophages with an anti-inflammatory phenotype[42]. Indeed, the frequency of mucosal vaccine-induced $CD73^+$ M2-like macrophages correlated with that of $CD14^+$ efferocytes measured in blood at week 13 ($p = 0.038/\rho = 0.61$; Fig. 3b), strengthening the role of efferocytes in shaping the mucosal pro-resolving macrophages. Moreover, their frequency correlated with the antibody titer targeting $\Delta V1gp120$ measured in the vaginal secretions collected at weeks 14–16 ($p = 0.012/\rho = 0.71$; Supplementary Fig. 4c). In addition to efferocytosis, tolerogenic DC-10 can also support anti-inflammatory

response[43], and in our prior study, the frequency of DC-10 in blood was correlated with efferocytosis and vaccine efficacy[28]. Comparison of mucosal vaccine-induced DC-10 frequencies among animals boosted with Alum or ALFQA showed no difference in their frequencies (Supplementary Figs. 3a, 4d). However, in the ALFQA group, vaccine-induced DC-10 significantly correlated with the number of challenges required for infection ($p = 0.049/\rho = 0.59$; Fig. 3c). The same correlation in the Alum group could not be tested because the animals used for comparison of mucosal responses were subsequently treated with microbicide during the challenge phase. These data suggest that although the ALFQA did not change the frequency of DC-10, it may have affected their function. However, additional studies are needed to confirm this hypothesis. Surprisingly, vaccination with ALFQA had an opposite effect on the expression of CD73 on DC-10 than in M2-like macrophages and induced lower $CD73^+$ DC-10 compared to vaccination with alum ($p = 0.024$; Supplementary Fig. 4e). Despite this, $CD73^+$ M2-like macrophages and $CD73^+$ DC-10 were positively correlated in both Alum and ALFQA animals (Supplementary Fig. 4f), suggesting crosstalk between these myeloid populations in the context of both vaccines, however additional studies would be required to confirm this hypothesis. These data point towards the ability of the ALFQA adjuvant to promote the induction of systemic and mucosal pro-resolving anti-inflammatory immune responses that promote efficacy.

## ALFQA HIV $\Delta V1gp120$ boost increases mucosal $CD73^+CD163^+$ anti-inflammatory macrophages in macaques

To confirm and expand on our findings at mucosal sites, we performed another study with twelve macaques immunized with identical vaccine platforms but based on HIV immunogens that will be used in the CLEAR trial (Supplementary Fig. 4g). In this study, animals were primed twice with DNA-HIV plasmids expressing clade AE-A244 $\Delta V1gp160$ env and clade B-HXB2 p55gag. Animals were then boosted with ALVAC-HIV (vCP2438) expressing Clade B-IIB gag-pro, clade C-ZM96 V1-replete gp120, and clade B-LAI gp120-transmembrane domain, alone, or together with clade AE-A244 $\Delta V1gp120$ protein adjuvanted in ALFQA ($n = 6$) or alum (Rehydragel; $n = 6$). Mucosal samples were collected 1 week following protein boost (week 13), but not at baseline. The analysis identified that, following HIV-based DNA/ALVAC/gp120 vaccination, the frequency of $CD73^+CD163^+$ M2-like macrophages was significantly higher in animals immunized with the protein boost formulated in ALFQA than in those with alum ($p = 0.026$; Fig. 3d). These data confirmed the ability of ALFQA to induce tolerogenic mucosal responses when used to formulate both SIV and HIV protein boosts. Therefore, the use of ALFQA to formulate anti-HIV vaccines for humans could result in the induction of mucosal pro-resolving responses beneficial for vaccine efficacy.

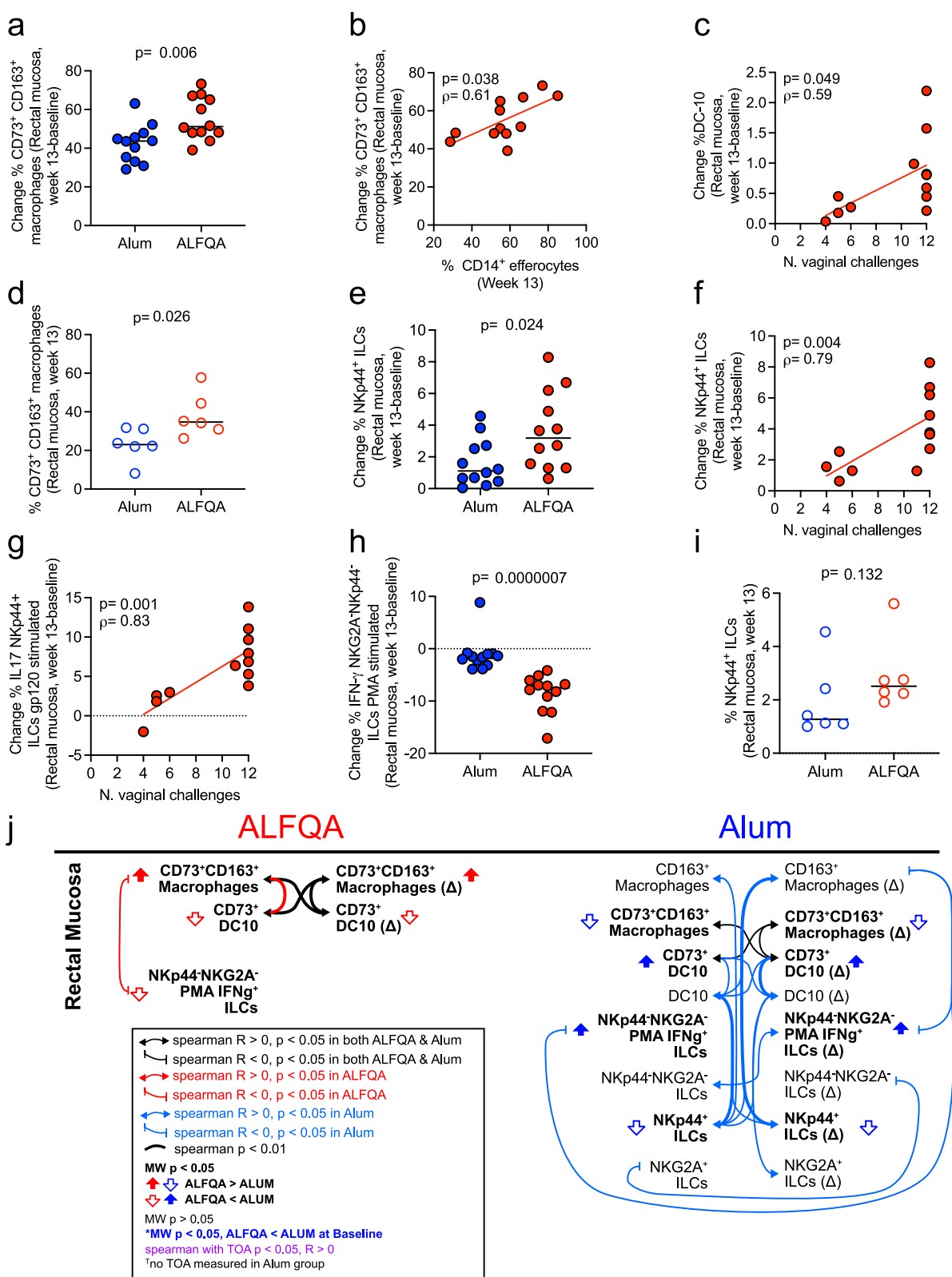

**ALFQA augments mucosal NKp44+ innate lymphoid cells and decreases mucosal interferon-γ-producing NKG2A−NKp44− cells**

In our previous studies, we identified that rectal innate lymphoid cells (ILCs), classified based on the expression of NKG2A and NKp44 receptors, have opposite effects on vaccine efficacy. NKp44+ ILCs that produce IL-17 following stimulation with peptides encompassing the env sequence of SIV had a protective effect, whereas Phorbol 12-

myristate 13-acetate (PMA) -stimulated NKG2A−NKp44− ILCs that produce interferon-γ (IFN-γ) had a detrimental effect on vaccine efficacy[32,44]. Thus, to test the effect of ALFQA on mucosal ILCs we analyzed by flow cytometry their frequencies in the rectal biopsies. Here, rectal mucosa samples collected at baseline and 1 week following protein boost formulated in ALFQA or Alum were analyzed and compared. The phenotypic analysis of ILCs (Supplementary Fig. 3a)

**Fig. 3 | Mucosal vaccine-induced immune responses. a** Vaccine-induced change (week 13 – baseline) of the frequency of mucosal CD73⁺CD163⁺ macrophages in rectum of Alum ($n = 12$) and ALFQA ($n = 12$) animals. **b** Correlation between vaccine-induced change (week 13 – baseline) of rectal CD73⁺CD163⁺ macrophages and the frequency of CD14⁺ efferocytes at week 13 in ALFQA ($n = 12$) animals. **c** Correlation of the vaccine-induced change (week 13 – baseline) of the frequency of mucosal DC-10 in rectum and the time of acquisition (TOA) in ALFQA ($n = 12$) animals. **d** Frequency of mucosal CD73⁺CD163⁺ macrophages in rectum of Alum ($n = 6$) and ALFQA ($n = 6$) HIV-vaccinated animals (week 13). **e** Vaccine-induced change (week 13 – baseline) of the frequency of mucosal NKp44⁺ innate lymphoid cells (ILCs) in rectum of Alum ($n = 12$) and ALFQA ($n = 12$) animals. **f** Correlation of the vaccine-induced change (week 13 – baseline) of the frequency of NKp44⁺ ILCs in rectum and TOA in ALFQA ($n = 12$) animals. **g** Correlation of the vaccine-induced change (week 13 – baseline) of the frequency of rectal IL-17-producing NKp44⁺ ILCs following stimulation with gp120 protein and TOA in ALFQA ($n = 12$) animals. **h** Vaccine-induced change (week 13 – baseline) of the frequency of rectal IFN-γ-producing NKG2A⁻NKp44⁻ ILCs following stimulation with PMA of Alum ($n = 12$) and ALFQA ($n = 12$) animals. **i** Frequency of mucosal NKp44⁺ ILCs in rectum of Alum ($n = 6$) and ALFQA ($n = 6$) HIV-vaccinated animals (week 13). **j** Schematic summarizing Spearman correlations among mucosal cell populations in ALFQA animals (left) and alum (right) animals. Associations of $p < 0.05$ are shown with a thin line, and $p < 0.01$ with a thicker line connecting variables. Associations found in both Alum and ALFQA are shown in black. Associations found only in ALFQA or Alum animals are in red or blue, respectively. Double-headed arrows indicate Spearman $R > 0$, vertical stubs indicate Spearman $R < 0$. Two-tailed Mann–Whitney $p < 0.05$ direction between groups is indicated by the vertical block arrows. Positive Spearman correlation with TOA $p < 0.05$ is depicted with purple text. Comparisons: **a**, **d**, **e**, **h**, **i**, two-tailed Mann–Whitney $U$- test with median; Correlations: **b**, **c**, **f**, **g**, two-tailed Spearman correlation with simple linear regression. Alum animals are depicted in blue, ALFQA animals are depicted in red. In **a**–**c**, **e**–**h**, SIV-immunized alum and ALFQA animals are depicted as black circles filled in blue and red, respectively. In **d**, **i**, HIV-immunized Alum and ALFQA animals are depicted as empty circles with blue and red borders, respectively. Source data are provided as a Source Data file.

identified a significant difference between ALFQA and alum groups in the vaccine-induced frequencies of ILCs expressing only NKp44⁺ ($p = 0.024$; Fig. 3e), but not in those expressing only NKG2A or double negative for both receptors (Supplementary Fig. 4h, i). Interestingly, in the ALFQA vaccinated animals, the frequency of vaccine-induced mucosal NKp44⁺ cells strongly correlated with vaccine efficacy ($p = 0.004/\rho = 0.79$; Fig. 3f). We then investigated the ability of ALFQA to influence the production of IFN-γ and IL-17 cytokines, by mucosal ILCs stimulated with either peptides encompassing the whole gp120 protein or PMA. Following gp120 stimulation, no difference was seen between alum and ALFQA groups, but in the ALFQA group, the vaccine-induced change in the frequency of NKp44⁺ ILCs expressing IL-17 strongly correlated with decreased risk of acquisition ($p = 0.001/\rho = 0.83$; Fig. 3g), confirming the importance of these cells in the efficacy of Alum vaccination[32]. Remarkably, compared to Alum, ALFQA induced fewer pro-inflammatory mucosal PMA-stimulated NKG2A⁻NKp44⁻ ILCs producing IFN-γ ($p < 0.001$; Fig. 3h), which have been associated with an increased risk of virus acquisition in a previous study[32].

These data suggest that the protein boost formulated in ALFQA leverages the effect already mediated by alum. On the one hand, the boosts promote the induction of beneficial NKp44⁺ ILCs that help in maintaining mucosal homeostasis, while, on the other, they decrease the detrimental pro-inflammatory ILCs. Similarly, we also assessed the effect of ALFQA on mucosal ILCs in animals administered the HIV-based CLEAR immunogens, as done for tolerogenic macrophages and DC-10. Following anti-HIV ΔV1DNA/ALVAC/ΔV1gp120 vaccination (week 13), the ALFQA group showed trends towards increased frequency of mucosal NKp44⁺ ILCs ($p = 0.132$; Fig. 3i) and partially decreased frequency of mucosal IFN-γ-producing NKG2A⁻NKp44⁻ ILCs ($p = 0.394$; Supplementary Fig. 4j) when compared to the Alum group, supporting the results obtained with the SIV-based vaccination.

## Crosstalk between vaccine-induced myeloid and innate lymphoid cells

To further investigate the crosstalk between mucosal cellular response, we examined the association of ILCs with myeloid populations in the rectal mucosa (Fig. 3j). In the Alum regimen, the frequency of DC-10 was positively associated with NKp44⁺ ILCs (Supplementary Fig. 4k), while the vaccine-induced change in CD163⁺ macrophages was positively associated with NKp44⁺ and negatively with IFN-γ-producing NKp44⁻NKG2A⁻ ILCs (Supplementary Fig. 4l, m). Strikingly, none of these relationships were found in the ALFQA vaccinated group, and instead, ALFQA induced a stronger correlation between mucosal CD73⁺ DC-10 and CD73⁺ M2-like macrophages (Supplementary Fig. 4f), which negatively correlated with IFN-γ-producing NKp44⁻NKG2A⁻ ILCs (Supplementary Fig. 4n). Finally, while paired samples were not available in the Alum group, ALFQA animals showed a strong positive

correlation between NKp44⁺ ILCs in the rectal mucosa and V2-specific ADCC killing in the plasma (Supplementary Fig. 4o). As in the case of systemic immune responses, these data suggest that ALFQA and alum induce qualitatively different crosstalk among mucosal immune cells.

## ALFQA-induced plasma proteome promotes innate cell activation and development

We next investigated the cytokine/chemokine milieu induced by vaccination with ALFQA and Alum by proximity extension assay (PEA), measuring absolute plasma levels (pg/ml) of 45 biomarkers. Proteomic analysis was conducted on samples collected at baseline, and at 24 hours (week 12 + 24 h) and 1 week (week 13) following the last immunization. Since not all 45 biomarkers were detectable in the two groups and at all timepoints, only the 36 detectable biomarkers were considered for analyses. First, we analyzed the overall differences between ALFQA and Alum groups by principal component analysis. We did not identify a remarkable difference in plasma proteome at baseline (Supplementary Fig. 5a), whereas at week 12 + 24 h (Fig. 4a) and week 13 (Fig. 4b) the groups clearly separated in the first two principal components, indicating a divergent effect of the adjuvants on the proteome. Mann–Whitney tests between the levels of biomarkers measured in ALFQA and Alum immunized animals identified significantly different biomarkers at each timepoint. Unexpectedly, four biomarkers differed between the Alum and ALFQA groups at baseline. At this pre-experimental timepoint, the levels of IL-15 and C-C motif chemokine 11 (CCL11) were higher in the ALFQA group, whereas the levels of tumor necrosis factor (TNF) and CXCL8 were higher in the alum group (Supplementary Fig. 5b). While IL-15 remained higher in ALFQA animals at both subsequent timepoints, CCL11 levels became indistinguishable between groups at week 12 + 24 h and lower in ALFQA by week 13 (Supplementary Fig. 5c). CXCL8 levels remained higher in alum at week 12 + 24 h, but reached comparable levels in both at week 13, while TNF levels began higher in alum, flipped to higher concentration in ALFQA at week 12 + 24 h, and then returned to be higher in alum at week 13 (Supplementary Fig. 5c). Following vaccination, at week 12 + 24 h, ALFQA had higher levels of CCL4, colony stimulating factor 1 (CSF1), CXCL10, IL-18, LTA, CCL3, CCL19, CCL8, CSF2, CXCL9, Fms-related tyrosine kinase 3 ligand (FLT3LG), IL-17A, IL-33, TNF superfamily member 10 (TNFSF-10), and CCL2, with lower levels of matrix metallopeptidase 1 (MMP1), CSF3, CXCL12, epidermal growth factor (EGF), hepatocyte growth factor (HGF), and IL-7 relative to alum (Fig. 4c, Supplementary Fig. 5d, and Supplementary Table 4). At week 13, CCL4, CSF1, CXCL10, IL18, and LTA remained higher while MMP1 remained lower in the ALFQA group, while CCL2 flipped to lower in ALFQA animals at week 13 (Fig. 4d, e). ALFQA animals also displayed elevated TNFSF12 and vascular endothelial growth factor A (VEGFA) at week 13 compared with Alum (Fig. 4d, e, Supplementary Fig. 5e, and Supplementary Table 5). In prior studies, we found that

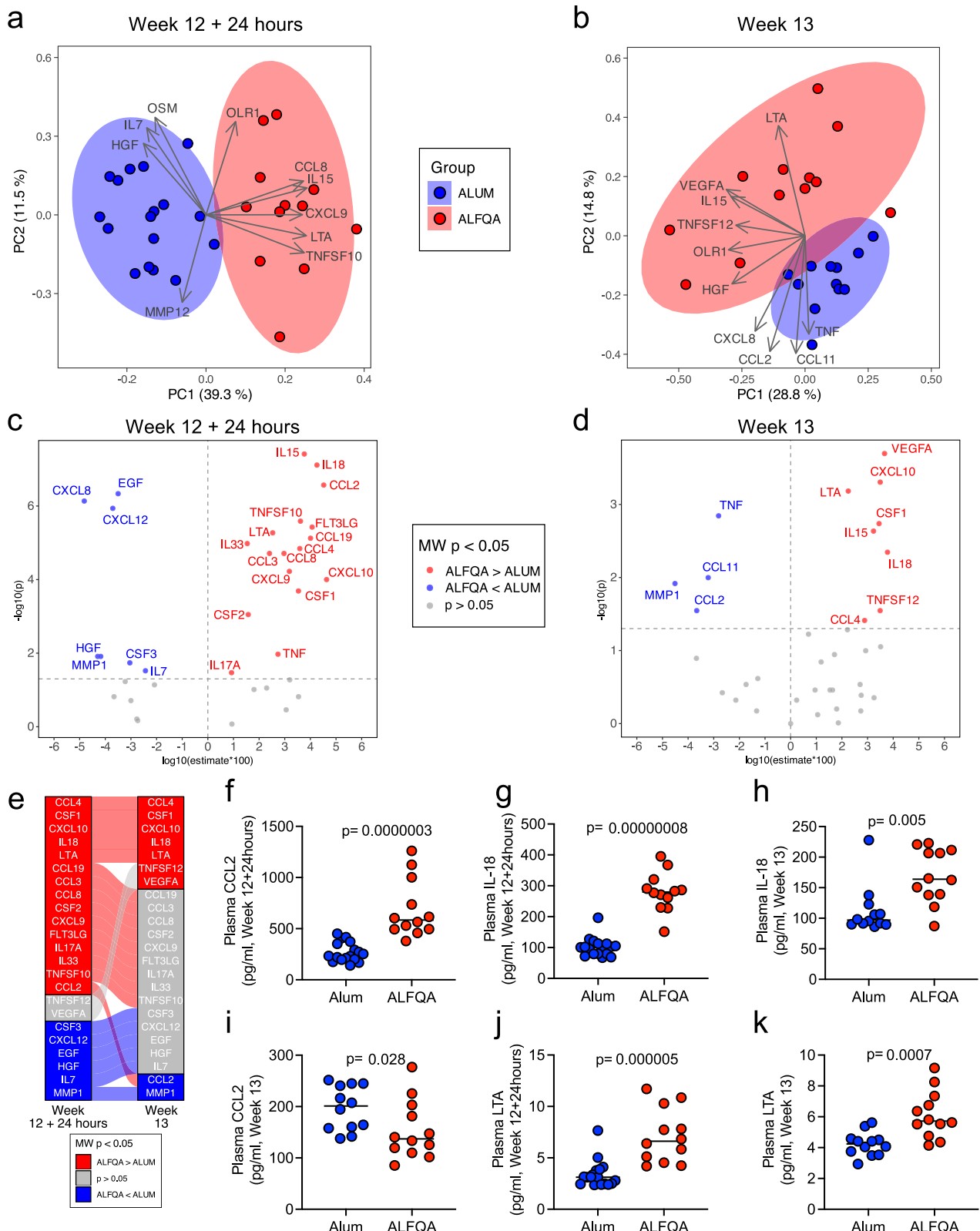

plasma CCL2 and IL-18 are important for engaging the CCL2/CCR2 axis in monocytes to maintain the crucial balance between pro- and anti-inflammatory responses and ultimately enable vaccine efficacy[26,28,45]. Indeed, when compared to Alum, the ALFQA group had higher levels of CCL2 in plasma at week 12 + 24 h ($p < 0.001$; Fig. 4f), as well as IL-18 at both weeks 12 + 24 h and 13 ($p < 0.001$, Fig. 4g; and $p = 0.005$, Fig. 4h), suggesting that ALFQA could favor this balance more than

alum and reduce the risk of virus acquisition, although the levels of CCL2 were higher at week 13 in the Alum group ($p = 0.028$; Fig. 4i). Lymphotoxin-α (LTA) is a member of the tumor necrosis factor superfamily, and can form a heterotrimer with lymphotoxin-β (LTβ), which regulates dendritic cells and CD4$^+$ T cell homeostasis[46]. Importantly, the LTA level was higher in ALFQA than in alum both at 1 and 7 days following protein boost ($p < 0.001$; Fig. 4j, k).

**Fig. 4 | ALFQA and alum-induced cytokine and chemokine milieu. a, b** Principal component analysis (PCA) of absolute levels (pg/ml) of cytokines, chemokines and other proteins measured in plasma collected at (**a**) week 12 + 24 h and **b** week 13 from alum (n = 17 at week 12 + 24 h and n = 12 at week 13) and ALFQA (n = 12 at both timepoints) animals. Arrows indicate the top drivers (loadings) for PC1 and PC2. **c, d** Volcano plots summarizing two-tailed Mann–Whitney differences in the plasma proteome between alum and ALFQA groups at **c** week 12 + 24 hs and **d** week 13 from Alum (n = 17 at week 12 + 24 h and n = 12 at week 13) and ALFQA (n = 12 at both timepoints) animals. The x-axis indicates the difference between the medians of each target level for each group, while the y-axis indicates the −log$_{10}$ (p values) of the ALFQA vs alum comparisons. Unadjusted p values are reported. The x-axis values are calculated as log$_{10}$ (absolute value (median of outer differences)) * sign of median of outer differences, as described in the methods. Only targets significantly different between the groups (p < 0.05) are labeled. Targets higher in the ALFQA group are marked in red, whereas targets higher in the Alum group are marked in blue. **e** Alluvial diagram summarizing the pattern of two-tailed Mann–Whitney significance (p < 0.05) over time. Targets that differ between the groups at baseline have been omitted. At each timepoint, targets are colored according to their direction of Mann–Whitney difference p < 0.05. Alluvial flow between the timepoints connect the same target at week 12 + 24 h and week 13 and are colored according to their pattern at week 12 + 24 h. (**f–k**) Absolute levels (pg/ml) of C-C motif chemokine ligand 2 (CCL2) at **f** week 12 + 24 h and **i** week 13, Interleukin 18 (IL-18) at **g** week 12 + 24 h and **h** week 13, and Lymphotoxin-alpha (LTA) at **j** week 12 + 24 h and **k** week 13, in plasma of Alum (n = 17 at week 12 + 24 h and n = 12 at week 13) and ALFQA (n = 12 at both timepoints). Comparisons: **f–k** two-tailed Mann–Whitney U-test with median. In panels **a, b** and **f–k**, SIV-immunized alum and ALFQA animals are depicted as black circles filled in blue and red, respectively. Source data are provided as a Source Data file.

To further probe the relationships among these cytokines/chemokines, we applied Spearman correlations within ALFQA or alum groups separately (Fig. 5a and Supplementary Data 1). We observed a strong positive correlation between CCL3 and CCL4 at baseline and week 12 + 24 h in both groups. Both CCL3 and CCL4 displayed higher levels in ALFQA than alum (week 12 + 24 h; Supplementary Table 4), and, at week 12 + 24 h, there was a strong association between CXCL9 and FLT3LG in the ALFQA group only, both of which were at increased levels relative to Alum at this timepoint (Supplementary Table 4). In contrast, in the alum group, CXCL9 correlated with a different network of chemokines, including CXCL11, CXCL10, CCL19, and CCL8. At week 12 + 24 h, while both groups displayed a strong relationship among CXCL8, IL-7, EGF, and HGF (which were all higher in the alum group; Supplementary Table 4). In the ALFQA group only, this network included OSM and remained strong by week 13, expanding to include CCL2 and VEGFA. In contrast, at week 13, alum animals had a strong direct association between CXCL8 and EGF.

We then sought to investigate which pathways may be activated by these distinct patterns using ingenuity pathway analysis (IPA, Qiagen). To focus on the contribution of the ALFQA adjuvant, IPA was done using the fold-changes of all 36 detectable targets between the ALFQA and alum groups. The IPA conducted at week 12 + 24 h identified several major biological themes and pathways (Supplementary Fig. 6a and Supplementary Data 2) induced by the ALFQA. Among these pathways were those involved in activation of monocytes, development of macrophages, induction and activation of dendritic cells, and maturation of phagocytes (Fig. 5b). At week 13, IPA identified the activation of the pathogen-induced cytokine storm signaling pathway, as well as induction of lymphocytes and the development of natural killer (NK) cells (Fig. 5c, Supplementary Fig. 6b, and Supplementary Data 3). The analysis, therefore, confirmed the ability of ALFQA to promote a plasma proteome favoring induction of the innate immune responses compared to Alum, as identified by flow cytometry and other canonical assays.

## DC-10 and plasma levels of LTA and CCL8 associated with decreased risk of SIV$_{mac251}$ acquisition

Finally, we performed correlation analyses between the levels of cytokines/chemokines, the number of challenges required for infection, and immune cell populations and functional activities. We identified a handful of biomarkers correlating with acquisition (Fig. 6a and Supplementary Fig. 7a). Interestingly, the level of LTA measured at week 13 correlated with a decreased risk of acquisition in ALFQA but not alum vaccinated animals (p = 0.007/ρ = 0.75 and p = 0.495/ρ = 0.22 respectively; Fig. 6b). LTA levels were also strongly associated with the frequency of vaccine-induced DC-10 in the mucosa (ALFQA animals, paired alum samples not available for analysis, p < 0.001/ρ = 0.90; Fig. 6c). Although it was not possible to test the association between plasma proteome and protective mucosal DC-10 responses in the Alum animals, in the ALFQA group changes in mucosal DC-10 at week

12 + 24 h were strongly positively associated with MMP12 and CCL13 levels and strongly negatively associated with VEGFA level, implicating additional early biomarkers that may favor DC-10 recruitment (Supplementary Fig. 7b–e). In the ALFQA group only, the levels of C-C motif chemokine ligand 8 measured at week 12 + 24 h correlated with lower risk of SIV acquisition (CCL8; ALFQA p = 0.043/ρ = 0.60; alum p = 0.986/ρ = -0.0051; Fig. 6d). At this same timepoint, we observed CCL8 to be positively associated with CCL2, CXCL9, and CXCL10 in both alum and ALFQA vaccinated animals, but only with CXCL12 in the ALFQA group and only with CXCL11, CCL19, CCL4, and CCL3 in alum (Supplementary Fig. 7f and Supplementary Data 1). Almost all of these biomarkers (except CXCL11) displayed higher levels in ALFQA compared with alum, suggesting that the CCL8 cytokine network is enhanced and takes on a distinct character in ALFQA vaccinated animals. Interestingly, CCL8 is highly expressed in M2-like macrophages, and it is induced by lactate in hypoxic niches and binds to CCR5[47], the coreceptor of HIV infection.

In the ALFQA group, none of the measured cytokines/chemokines correlated with increased risk (Fig. 6a). In contrast, in the alum group the levels of IL-17F (alum p = 0.041/ρ = -0.60; ALFQA p = 0.743/ρ = 0.11; Supplementary Fig. 8a), CXCL8 (alum p = 0.037/ρ = −0.61; ALFQA p = 0.707/ρ = −0.12; Supplementary Fig. 8b), and epidermal growth factor (EGF; alum p = 0.019/ρ = −0.68; ALFQA p = 0.771/ρ = −0.094; Supplementary Fig. 8c) measured at week 13 correlated with higher risk of SIV acquisition. While EGF and CXCL8 at week 13 were strongly correlated in both vaccine groups (alum p < 0.001/ρ = 0.87; ALFQA p = 0.007/ρ = 0.75; Supplementary Fig. 8d), this pair of biomarkers had a completely different biomarker interaction network in the ALFQA and alum-treated animals (Supplementary Fig. 8e and Supplementary Data 1). IL-17F positively associated with CXCL9 and CXCL10 at week 13 in ALFQA animals but was not associated with any biomarkers or immune phenotypes tested in alum animals.

Taken together, these data suggest that the ALFQA adjuvant orchestrates a combination of cytokines and chemokines that favors the development of immunomodulatory tolerogenic dendritic cells and tilts the balance to anti-inflammatory responses.

## Discussion

A decade of iterative macaque studies using a highly diverse pathogenic SIV$_{mac251}$ virus stock[48] have allowed us to substantially improve on the protective ALVAC-based/protein boost vaccine regimen with the introduction of DNA priming[28,45] and VLPs presenting ΔV1gp120 proteins engineered to expose the vulnerable helical V2 to antibody binding[27]. Nevertheless, further improvement of this approach is needed before it can be translated into a licensed HIV vaccine for human use. The use of ALF-based adjuvants in HIV vaccine formulation is relatively recent, and investigations on the viability of adjuvants in this family, particularly ALFQA, are still ongoing. Currently, the RV546 phase I clinical HIV vaccine trial underway in Thailand does include administration of full-length single chain (FLSC) gp120-CD4 chimera

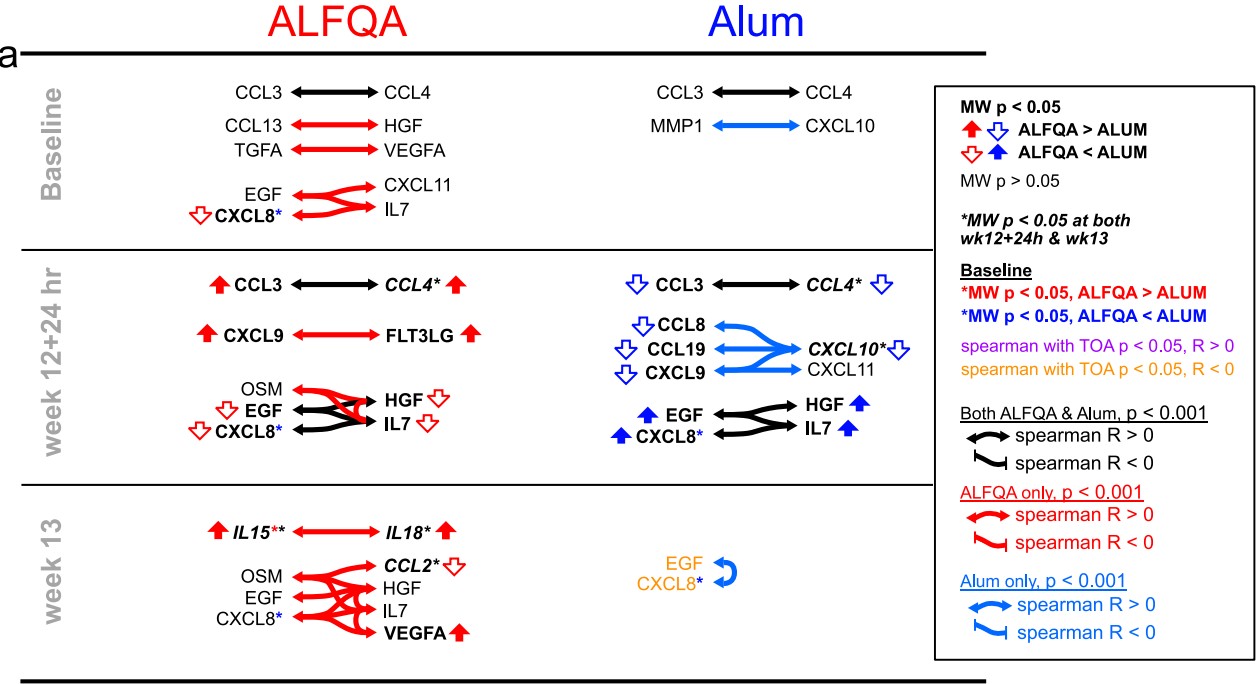

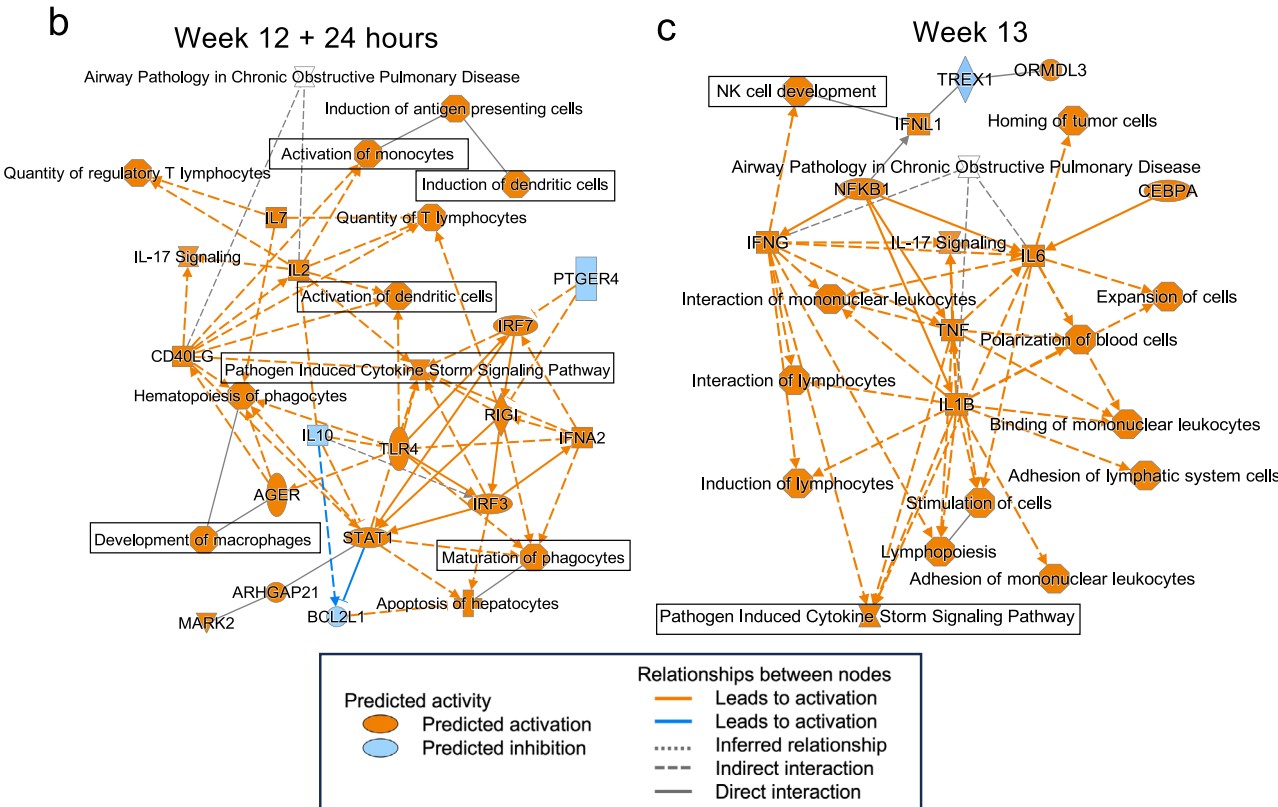

**Fig. 5 | Associations among biomarkers and ALFQA-induced proteome pathways. a** Schematic summarizing Spearman correlations among plasma proteome biomarkers in ALFQA (left) and alum (right) animals. Associations with $p < 0.001$ are indicated. Associations found in both Alum and ALFQA are shown in black. Associations found only in ALFQA or alum animals are in red or blue, respectively. Double-headed arrows indicate Spearman $R > 0$; there are no negative associations at this threshold. Two-tailed Mann–Whitney $p < 0.05$ direction between groups is indicated by vertical block arrows. Association with time of acquisition (TOA) $p < 0.05$ is depicted with purple text. **b**, **c** Graphical summaries of major biological themes identified by ingenuity pathway analysis (IPA) of ALFQA-induced proteome at **b** week 12 + 24 hours (alum $n = 17$ and ALFQA $n = 12$) and **c** week 13 (Alum $n = 12$ and ALFQA $n = 12$), compared to alum. Red and blue boxes indicate predicted activation and predicted inhibition, respectively; whereas red and blue lines indicate relationships that lead to activation and inhibition, respectively. Source data are provided as a Source Data file.

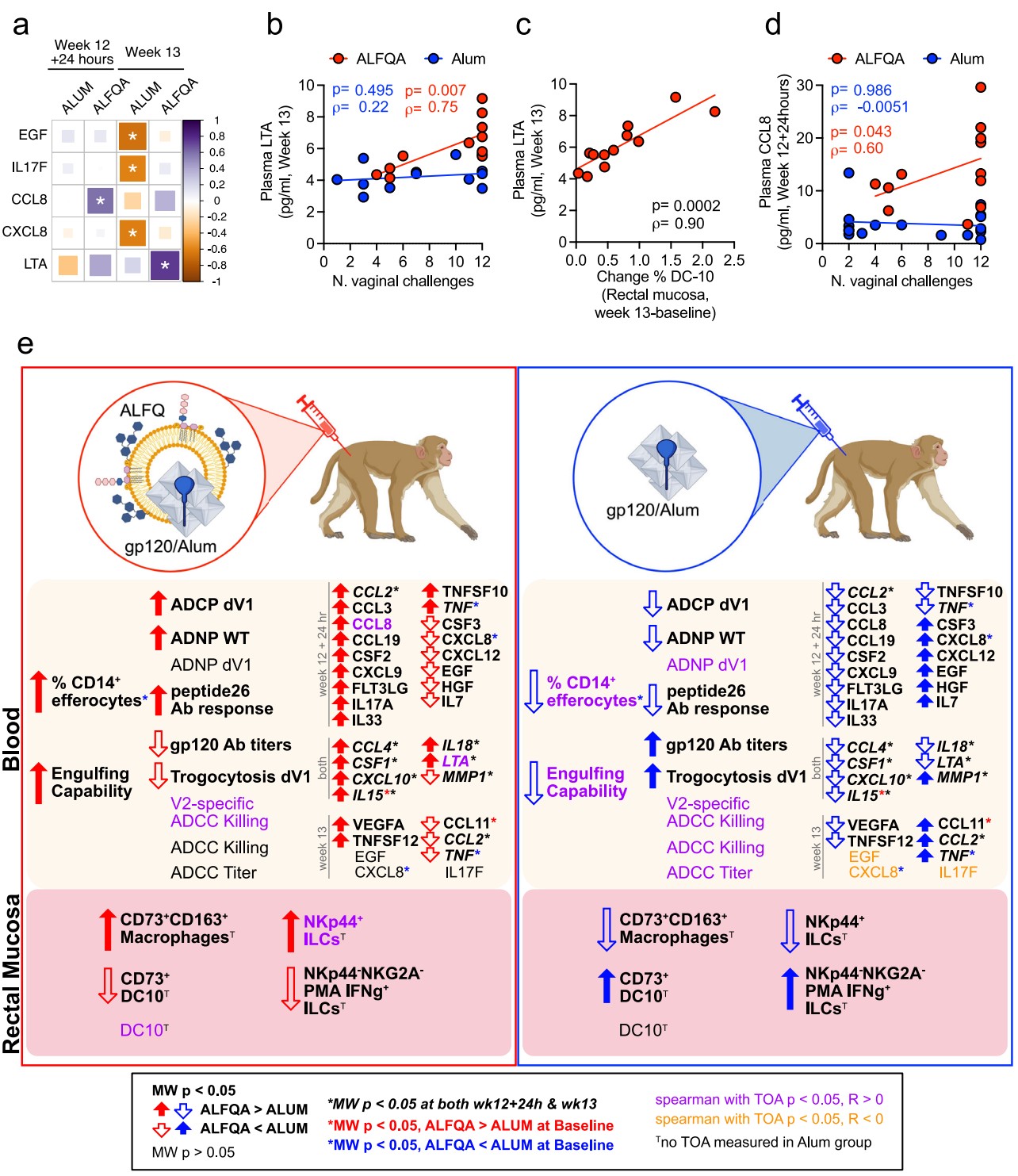

and A244 protein, both formulated in aluminum-based adjuvants, administered separately, but simultaneously, with ALFQ. However, no human study to date has tested the administration of HIV immunogens co-formulated with ALFQ and alum in the same preparation. Additional NHP studies which use ALFQA to deliver HIV vaccines are also ongoing and have not been published. Furthermore, the effect of ALFQA at the mucosal level, which is the primary site of HIV transmission and infection, has not been investigated at all.

For the first time, we used the $SIV_{mac251}$ macaque model to test the efficacy and immunogenicity of ALFQA as an adjuvant in a ΔV1DNA/ ALVAC/gp120 SIV/HIV vaccine candidate. We demonstrated that the addition of ALFQ to ΔV1gp120 protein formulated in Alum (ALFQA) not

only improves systemic responses that have been associated with decreased risk of $SIV_{mac251}$ in prior studies, such as antibodies to helical V2 and monocyte efferocytosis, but, importantly, that it also influences mucosal responses and immunity. ALFQA augments the frequency of CD73-expressing M2-like macrophages, which contribute to tissue repair and promote Th2 responses[49]. CD73 is a surface-expressed ecto-5'-nucleotidase that converts AMP (adenosine monophosphate) into adenosine, a potent regulator of inflammation[50,51], and suppressor of T-cell activation[52]. By increasing the frequency of these cells, ALFQA can therefore modulate the recruitment of target T cells. In addition, CD73 expression is induced by hypoxia via hypoxia-inducible factor-1[53], a transcription factor crucial for vaccine efficacy[45] that supports

**Fig. 6 | Targets associated with decreased risk and hypothetical mechanisms of vaccine efficacy. a** Heatmap summarizing the significant two-tailed Spearman correlations between the absolute levels (pg/ml) of epidermal growth factor (EGF), Interleukin 17 F (IL-17F), C-C motif chemokine ligand 8 (CCL8), C-X-C motif chemokine ligand 8 (CXCL8), and lymphotoxin-alpha (LTA) measured in plasma collected at week 12 + 24 h from ALFQA ($n = 12$) and alum ($n = 17$) animals, and at week 13 from ALFQA ($n = 12$) and alum ($n = 12$) animals, and time of acquisition (TOA). The ρ-values are identified by the color-scale, while the significant correlations ($p < 0.05$) are identified by asterisks. Reported $p$ values are unadjusted for multiple comparisons. **b, c** Correlation of the absolute level (pg/ml) of plasma LTA at week 13 with **b** TOA in ALFQA ($n = 12$) or alum ($n = 12$) animals or **c** the vaccine-induced change (week 13 – baseline) of the frequency of rectal DC-10 in ALFQA ($n = 12$) animals (paired data not available in alum-treated animals). **d** Correlations of the absolute level (pg/ml) of plasma CCL8 at week 12 + 24 h with TOA in alum ($n = 17$) and ALFQA ($n = 12$) animals. **e** Schematic representation of immune responses contributing to vaccine efficacy at the systemic and mucosal levels and their comparison between ALFQA (left) and Alum (right) groups. In peripheral blood, the immunization with ΔV1DNA/ALVAC and ΔV1gp120 protein boost adjuvanted in ALFQA induces antibodies to gp120 mediating ADCC, and, when compared to Alum, higher ADCP and ADNP, together with higher CD14⁺ efferocytes. In the ALFQA group, vaccination induces higher levels of LTA and other cytokines/chemokines. LTA in plasma induces mucosal tolerogenic DC-10, toward a protective phenotype. Together with increased anti-inflammatory CD73⁺CD163⁺ macrophages and ILCs that express NKp44⁺ and maintain tissue-homeostasis, DC-10 favors the killing and clearance of SIV-infected apoptotic cells, preventing the recruitment of CD4⁺ target T cells. The figure contains images (monkeys and immunogens) created in BioRender. Woode, E. (2025) https://BioRender.com/p4mqsnf. Correlations: **b–d** two-tailed Spearman correlation with simple linear regression. Alum animals and correlations are depicted in blue, ALFQA animals and correlations are depicted in red. Source data are provided as a Source Data file.

efferocytosis by suppressing multiple pro-inflammatory cytokines[42]. Indeed, the mucosal CD73⁺ M2-like macrophages correlated with the frequency of efferocytes in blood in this study, strengthening the relation between these two cells and suggesting that efferocytes may migrate to the mucosa. The multiple components of the ALFQA adjuvant, containing synthetic MPLA-like, 3D-PHAD, QS21, and Alum, are likely responsible for the increased recruitment of these cells when compared to Alum alone. Alum induces activation of the CCR2/CCL2 axis and recruitment of monocytes[28,54,55]. Alum, as well as QS21 co-administered with MPLA, elicit caspase-1-dependent IL-1β and IL-18 release in APCs, such as macrophages and dendritic cells, and NLRP3 inflammasome activation, components which induce and recruit monocytes and NK cells[56–58].

In addition to tolerogenic macrophages, our ALFQA-adjuvanted vaccine also affects mucosal DC-10, cells known to produce IL-10 and promote tolerance by inducing T regulatory type 1 CD4 T cells (Tr1)[43]. We showed that DC-10 frequency was associated with lymphotoxin-α production. LTA, a member of the tumor necrosis factor (TNF) superfamily, can form homotrimers or heterotrimers with LTβ, which bind to TNF receptors 1 and 2 and to lymphotoxin-β receptor (LTβR), respectively[46]. LTβR is expressed on several types of dendritic cells and is part of a complex signaling network that can have both positive and inhibitory effects on different DC subsets[59]. In the gut, LTβR activation induces the maturation of DCs producing inducible nitric oxide synthase (iNOS⁺ DCs)[46]. Indeed, extrinsic nitric oxide inhibits the differentiation of effector DCs cells with the potential to promote severe inflammatory responses in the intestine[60], therefore favoring the induction of tolerogenic DC-10. Since the levels of vaccine-induced DC-10 were not different between the two vaccine groups (Supplementary Fig. 4d), LTA may have effected mainly the quality and function of DC-10 rather than their frequency. Indeed, activation of LTA receptors on conventional dendritic cells can modulate their immune responses and change their ability to produce cytokines and polarize T cells[46,59]; however, its activity on DC-10 is unknown. Interestingly, and in contrast to the vaccine-induced CD73⁺ M2-like macrophages, the levels of vaccine-induced CD73-expressing DC-10 were lower in the ALFQA than in the Alum group. The role of the expression of CD73 in DC-10 is not well documented; however, lowered expression of CD73 impacts the migration of skin dendritic cells[61], suggesting that lower CD73 on DC-10 might have affected their levels at the mucosal site. Additional studies are required to elucidate the expression of CD73 in DC-10, its possible impact on their migration and function, as well as the role of LTA on their functionality, to fully understand the protective role of DC-10 with this vaccine modality.

Here, we identified the important role of different innate immune responses in mediating a reduced risk of SIV acquisition. Although innate immunity may be short-lived, potentially limiting the duration of the vaccine efficacy, our prior studies showed the memory properties of NK cells and ILCs generated by this vaccine regimen[62] as well as epigenetic reprogramming of monocytes[28], which might contribute to extending vaccine efficacy. Further studies will be needed to evaluate the long-term durability of these innate mucosal immune responses.

In summary (Fig. 6e), immunization with ΔV1DNA/ALVAC/gp120 vaccine adjuvanted with ALFQA rather than Alum results in increased secretion of LTA, CCL8, CCL2, and IL-18, and other chemokines/cytokines. This favored the induction and recruitment of tolerogenic DC-10, anti-inflammatory macrophages (CD163⁺CD73⁺), and NKp44 ILCs in the mucosa. Together, these tolerogenic cells conduct more efficient efferocytosis and favor the non-inflammatory elimination of apoptotic, infected cells killed by ADCC, but also ADCP, and ADNP, which were both increased by the ALFQA vaccine strategy. NKp44⁺ ILCs further complements these immune effects by maintaining mucosal integrity and cell homeostasis, and potentially contributing their own V2-specific cytotoxicity, therefore reducing the recruitment of pro-inflammatory CD4⁺ T cells that could provide additional target cells for the virus.

## Limitations of the study
The SIV immunogenicity/efficacy study was not designed to compare vaccine efficacy between macaques vaccinated with the protein boost formulated in alum or that formulated in ALFQA, but rather to compare each vaccinated group to naïve controls. The HIV immunogenicity study included a limited number of macaques ($n = 6$/group).

## Methods
### Ethics statement
The research reported in the manuscript complies with all National Institutes of Health ethical regulations, and it was approved by the Center for Cancer Research non-human primate animal study protocol prospective scientific committee.

### Animals, vaccines, and SIV_mac251 challenge
The animals enrolled in the study were Indian rhesus macaques (*Macaca mulatta*). Macaques were provided by Alpha Genesis Inc. (Yemasee, SC) and Primate Products Inc. (Immokalee, FL) and were housed at the National Institutes of Health (Bethesda, MD) and handled in accordance with the standards of the Association for the Assessment and Accreditation of Laboratory Animal Care (AAALAC) in an AAALAC-accredited facility (OLAW, Animal Welfare Assurance A4149-01). Animal care and procedures were performed under animal study protocols approved by the NCI Animal Care and Use Committees (ACUC; Protocol numbers: VB-013, VB026, VB034, VB042, and VB047). Animals were monitored daily for any signs of illness, and appropriate medical care was provided as needed. Animals were socially housed per the approved ACUC protocol and social compatibility, except during the viral challenge phase, when they were individually housed. All clinical procedures, including biopsy collection, administration of

anesthetics and analgesics, and euthanasia, were conducted under the direction of a laboratory animal veterinarian.

### V1-deleted DNA/ALVAC/gp120/ ALFQA SIV vaccination study

Twelve female macaques with an average age of 4.17 years (Standard deviation 0.51) were included in the ALFQA group. The animals were intramuscularly immunized twice (weeks 0 and 4) with DNA constructs 206S SIV p57gag$_{mac239}$ (1 mg/dose) and V1-deleted SIV$_{mac251-M766}$ gp160$_{\Delta V1}$ (2 mg/dose), mixed together and administered concomitantly in the two thighs (half-dose per thigh), as previously done[28]. All twelve macaques were then boosted twice (weeks 8 and 12) with intramuscular inoculations of $10^8$ plaque-forming units (PFU) of recombinant ALVAC (vCP2432), expressing SIV$_{mac251}$ *gag-pro* and *gp120TM* (Sanofi Pasteur, Bridgewater, NJ). During the second ALVAC boost (week 12) macaques were also administered 400 µg of V1-deleted SIV$_{mac251-M766}$ gp120$_{\Delta V1}$, adjuvanted in ALFQA (Fig. 1a, ALFQA group). The protein boost was formulated by adsorbing 400 µg SIV$_{mac251-M766}$ gp120$_{\Delta V1}$ protein diluted in phosphate-buffered saline (PBS) to 850 µg aluminum Al$^{3+}$ (as aluminum hydroxide fluid gel suspension Rehydragel) for 10 min at room temperature (RT), and on a shaker. The protein adsorbed to rehydragel was then mixed with 200 µg of 3D-PHAD and 100 µg of QS21 and incubated on a shaker for another 10 min at RT. At the time of administration, the vaccine was briefly shaken, loaded into the syringe and promptly injected into the contralateral thigh of each animal.

During the immunization protocol, mucosal rectal biopsies were collected at baseline and 1 week following the ALVAC/protein boost (week 13). At the end of immunization (week 17), the animals were intravaginally challenged with 11 low doses of pathogenic SIV$_{mac251}$. Challenges were performed using a dilution of 1:25 of a stock of SIV$_{mac251}$ propagated in macaque cells (QBI#305342b, Quality Biological, Gaithersburg, MD), repeated once a week until confirmation of the infection by viral load performed in the plasma, and by administration of 1 mL of SIV$_{mac251}$ diluted in RPMI 1640 (Gibco, Waltham, MA) to a final concentration of 4000 TCID$_{50}$/mL (evaluated in rhesus 221 cells). Thirty-seven naïve rhesus macaques enrolled in prior studies[28], that were exposed to intravaginal SIV$_{mac251}$ challenges, following the same procedures, and using the same viral stock and at the same dilution as described above, were used as historical naïve controls.

### V1-deleted DNA/ALVAC/gp120/alum SIV vaccination study.

Viral acquisitions and immunological data obtained from thirty female macaques with an average age of 3.24 years (standard deviation 0.61) and immunized in prior studies[28,29] were used for comparison (Fig. 1a, alum group). Briefly, thirty female macaques received the same DNA primes (week 0 and 4), and ALVAC boosts (week 8 and 12) administered to the ALFQA group described above. At week 12, the protein boost, consisting of the same SIV$_{mac251-M766}$ gp120$_{\Delta V1}$ protein, was formulated by adsorbing the protein diluted in PBS to 5000 µg aluminum Al$^{3+}$ (as alhydrogel adjuvant 2%, Invivogen) for 10 min on a shaker at RT. Five weeks following the last vaccination, the animals were exposed to intravaginal SIV$_{mac251}$ challenges, following the same procedures, and using the same viral stock and at the same dilution, as described for the ALFQA group above.

### V1-deleted DNA/ALVAC/gp120/Alum SIV vaccination study for mucosal samples.

Mucosal samples from the rectum were obtained from twelve female macaques with an average age of 3.44 years (standard deviation 1.35) (Supplementary Fig. 4a) and immunized following the exact same vaccination regimen described above for the V1-deleted DNA/ALVAC/gp120/Alum SIV vaccination study. Mucosal samples were collected at baseline and 1 week following the protein boost adjuvanted in Rehydragel, following the same procedures and timeline used for the ALFQA group described above in the V1-deleted DNA/ALVAC/gp120/ ALFQA SIV vaccination study. Rectal mucosa

samples were processed as described below. Following the collection of mucosal samples, these twelve animals received unrelated treatments (microbicide), that were not performed in the ALFQA group. However, since the samples were collected before the treatments, it was possible to perform immunological comparisons between the alum and ALFQA groups.

### V1-deleted DNA/ALVAC/gp120/Alum and ALFQA HIV vaccination study for mucosal samples.

Mucosal samples from the rectum to compare alum and ALFQA groups immunized with HIV immunogens were obtained from twelve macaques with an average age of 3.32 years (standard deviation 0.31) (Supplementary Fig. 4g). Six males and six females, macaques were equally randomized in two groups. All 12 animals were intramuscularly immunized twice (weeks 0 and 4) with DNA constructs clade B-HXB2 p55$^{gag}$ (1 mg/dose) and clade AE-A244 V1-deleted ($\Delta$V1) gp120 (2 mg/dose), mixed together and administered. All macaques were then boosted twice (weeks 8 and 12) with intramuscular inoculations of $10^8$ plaque-forming units (PFU) of recombinant ALVAC (vCP2438), expressing Clade B-IIB gag-pro, clade C ZM96 V1-replete gp120, and clade B-LAI gp120-transmembrane domain (Sanofi Pasteur, Bridgewater, NJ). During the second ALVAC boost (week 12), macaques were also administered 400 µg of clade AE-A244 $\Delta$V1gp120 protein. In six macaques, the protein was adjuvanted with alum rehydragel (Supplementary Fig. 4g, alum group) and the other six with ALFQA (Supplementary Fig. 4g, ALFQA group). Formulation of the immunogens and their administration was performed as described for the SIV vaccination study. Mucosal rectal biopsies were concomitantly collected 1 week following the ALVAC/protein boost (week 13) and then processed as described below.

### Viral RNA load

The RNA copies of SIV$_{mac251}$ in plasma were quantified by digital droplet polymerase chain reaction (ddPCR) as in prior studies[45]. Briefly, following extraction, nucleic acid samples are prepared and mixed with primers and fluorescent probes, and Supermix (One-Step RT-ddPCR Advanced Kit; Bio-Rad Catalog No. 1864022). The following primers were used: SIV gag forward primer: 5′-GCAGAGGAGGAAATTACCCAGTAC-3′, SIV gag reverse primer: 5′-CAATTTTACCCAGGCATTTAATGTT-3′. The system partitions 20 µL of the RNA extract from plasma into 20,000 nL droplets. Each droplet contained a random distribution of the target and/or background RNA. The inactive reverse transcriptase and Taq DNA polymerase (Bio-Rad one-step ddPCR kit) blend into the droplets and activate due to the temperature increase at 50 °C. In each droplet, double-quenched SIV gag probe: 5′-/56FAM/TGTCCACCTGCCATTAAGCCCGA-3′ were included at the start of the reaction. The plate is then transferred to a beta-prototype droplet reader (optical reader). Poisson statistics are used to quantify the proportion of positive droplets, i.e., the number of target templates from which absolute viral RNA levels can be calculated precisely in copies/µL. Based on fluorescence amplitude, each 1 nL droplet is defined as either positive or negative. The fraction of fluorescent droplets determines the concentration of the target in the sample. Copies/mL were calculated based on copies/µL (Volume of mastermix per well/Volume of template) * plasma dilution factor/(volume of plasma used for extraction/elution volume). The assay's limit of quantification (LOQ) is set at 50 RNA copies per milliliter of plasma.

### Immunoglobulin G plasma titers to gp120

gp120 total IgG antibodies (immunoglobulin G) were measured by ELISA as previously described[27]. Briefly, ELISA plates were incubated overnight at 4 °C with 50 ng of SIV$_{mac251-M766}$ gp120 protein/well in 100 µl of 50 mM sodium bicarbonate buffer (pH 9.6). Following coating with the protein, the plates were washed and blocked for 1 h at

room temperature (RT) with 200 µl/well PBS Superblock (Thermo Fisher Scientific). Cryopreserved plasma collected at 3 weeks following last immunization (week 15), were thawed and serially diluted with sample diluent (Avioq), added to plates (100 µl/well), and incubated for 1 h at 37 °C. plates were then washed and incubated for 1 h at 37 °C with Horseradish Peroxidase HRP conjugated anti-human antibody (100 µl/well, diluted at 1:120,000 in sample diluent, Avioq). Finally, the plates were washed and developed using K-Blue Aqueous substrate and 2 N Sulfuric acid to block the reaction. Plates were read at 450 nm by using a Molecular Devices E-max plate reader. The titers were calculated as the highest dilution that provided an optical density (OD) value that was double the average value obtained for unvaccinated normal rhesus macaques.

### Pepscan

Plasma samples were assayed by PEPSCAN analysis using $SIV_{mac251}$ gp120 linear 20-mer peptides as previously described[27]. Cryopreserved plasma samples collected between 2 and 5 weeks following the last immunization (weeks 14–17) were analyzed. Briefly, ELISA plates were incubated overnight at 4 °C with 1000 ng/well of each of the V1 and V2 overlapping peptides (Supplementary Table 1) in 50 mM sodium bicarbonate buffer (pH 9.6). Following incubation, plates were blocked for 1 h at RT with Pierce SuperBlock blocking buffer, loaded with 100 µl/well diluted plasma (1:50 in sample diluent, Avioq), incubated for 1 h at 37 °C, washed, and incubated for 1 h at 37 °C with 100 µl/well anti-human HRP (diluted at 1:120,000 in sample diluent Avioq). The plates were then washed again and developed using 100 µl/well of K-Blue Aqueous substrate (Neogen) and incubating 30 min at RT. Two Normal Sulfuric acid (100 µl/well) was added to the plates to stop the reaction. The plate was read at 450 nm with a Molecular Devices E-max plate reader, and optical density for each sample was used for the analysis.

### Antibody avidity

Antibody avidity determinations were conducted using the Biacore 4000 surface plasmon resonance (SPR) system as previously described[63–66]. Briefly, the immobilizations were performed using a standard amine-coupling kit. The CM5 sensor chip (Cytiva) surface was activated with a 1:1 mixture of 0.4 M 1-ethyl-3-(3-dimethylaminopropyl) carbodiimide hydrochloride (EDC) and 0.1 M $N$-hydroxysuccinimide (NHS) (Cytiva) for 600 s Protein SIVmac251-M766-ΔV1gp120 (20 µg/mL, spot 1, 2, 4, and 5 in flow cell 1), and Streptavidin (1 µM, spot 1, 2, 4, and 5 in Flow cells 2, 3, and 4) in 10 mM sodium acetate pH 4.5 were immobilized on the CM5 sensor chip. Spot 3 of each flow cell was left unmodified to serve as a reference. The resulting Response Units (RUs) for flow cells 1, 2, 3, and 4, respectively, were as follows: 12542-14330 RU; 7807-8015 RU; 7060-7924 RU; and 7073-7920 RU. Biotinylated cyclic V2 peptide was injected onto the streptavidin immobilized surface (flow cells 2, 3, and 4) for capturing, and the contact time was 240 s. Capture level (RU) of biotinylated cV2 peptide ($SIV_{mac251}$) was as follows: 1951-2051 RU. Following the surface preparation, heat-inactivated (56 °C for 45 min) plasma samples were diluted 1:50 in running buffer (10 mM HEPES, 150 mM NaCl, 0.05% Tween-20, pH 7.4) and injected onto the protein, and peptide immobilized surface for 250–300 s followed by dissociation for 1300–3000 s. Data for each sample were collected at a rate of 10 Hz, with an analysis temperature of 25 °C. All sample injections were conducted at a flow rate of 10 µL/min. The bound surface was regenerated with 150 mM HCl for 60 s. Data analysis was performed using Biacore 4000 Evaluation software 4.1 with double subtractions for the unmodified surface and buffer blank. Fitting was conducted using the dissociation mode integrated with Evaluation software 4.1. The data were further processed using Microsoft excel (version 16.79.1), and GraphPad Prism (version 10.0.1). The data were shown as an avidity score. Avidity score was calculated as RU/ $K_d$.

### Plasma neutralizing antibodies

The levels of neutralizing antibodies were measured in the plasma of vaccinated animals collected between 2 to 3 weeks following the last immunization as a reduction in luciferase reporter gene expression after a single round of infection in TZM-bl cells. Test samples were serial-diluted (threefold dilution in duplicate) and incubated with 200 $TCID_{50}$ of virus in a total volume of 150 µl for 1 h at 37 °C in 96-well flat-bottom culture plates. TZM-bl cells were trypsinized and added to each well (10,000 cells in 100 µl of growth medium containing 20 µg/mL DEAE dextran). A set of wells with cells only was used as background control, and another set with cells and virus was used as virus control. After incubation (48 h), the cells were lysed by the addition of Britelite (PerkinElmer Life Sciences, Waltham, MA), and three quarters of the cell lysate were transferred to a 96-well black solid plate (Corning Costar) for luminescence measurement. Neutralization titers are defined as the dilution at which relative luminescence units were reduced by 50% ($ID_{50}$) or 80% ($ID_{80}$) compared to that in virus control wells after subtraction of background relative luminescence units. Neutralization was tested against the virus $SIV_{mac251.6}$ (ID #10848), $SIV_{smE660/BR-CG7G.IR1}$ (ID #1370DB2), $SIV_{smE660/BR-CG7G.IR1}$ (ID #1634DB2), and $SIV_{mac251}$ (challenge virus).

### gp120-specific IgG antibodies in vaginal secretions

gp120-specific IgG antibodies in vaginal secretions were measured following extraction from the swabs, followed by ELISA.

**Vaginal secretion extraction.** Extraction Buffer (0.8% sodium chloride, 0.67% Proclin 300 (Sigma, 48912-U), 1% Protease Inhibitor Cocktail (Calbiochem, 539131) in 1x Dulbecco's phosphate-buffered saline [DPBS]) was prepared and filtered with a 0.22 µm filter unit and stored at 4 °C. Frozen vaginal swabs were thawed at room temperature for 5 min and then placed on ice to completely thaw. Each swab was examined for visible blood. Extraction buffer was added to each tube containing a swab, and the mucosal solution was manually extracted by squeezing the swab with a pipette tip. Tubes were vortexed three times each for 1-min intervals prior to incubation at room temperature for 15 min. The swabs were centrifuged at 4 °C, 16,000×$g$ for 15 min. Supernatant was extracted and replicates were combined. The protein concentration of the mucosal extract was determined by Nanodrop at an absorbance of 280 nm.

**Antigen-specific IgG ELISAs.** ELISA titers were determined by coating plates with 0.1 µg/well coating antigen (ΔV1 $SIV_{M766}$ gp120) in 1X DPBS w/o Ca and Mg. Coated plates were incubated overnight at 4 °C. Plates were washed with Wash buffer (0.1% Tween-20 in 1XDPBS), and blocking buffer (5% milk, 0.1% Tween-20 in DPBS) was added to the plates and incubated at room temperature for 2 h. Vaginal mucosal extract was thawed at 4 °C, then diluted to an initial dilution of 1:2.5 in blocking buffer. An NHP sample from a previous study served as a positive control for specific IgG titers. Samples were added to the plate in triplicate and then serially diluted 1:2 down the plate, mixing thoroughly with each dilution. Plates containing the sample were incubated at room temperature for 2 h. Secondary antibody (goat anti-monkey IgG (gamma chain) HRP conjugated, Alpha Diagnostic, 70021) was diluted 1:2000 in blocking buffer. Plates were washed, secondary Ab was added to each plate and incubated at room temperature for 1 h. The plates were washed and substrate [KPL ABTS Peroxidase Soln. A (Fisher Scientific, Cat: 5120-0035), and Peroxidase Soln. B (Fisher Scientific, Cat: 5120-0038) were mixed and added to the plates. The plates were incubated in the dark for 1 h at RT. Stop solution (1% sodium dodecyl sulfate (Invitrogen, 24730-020) in diH2O) was added to each plate, and the plates were read on a Molecular Devices SpectraMax M5E plate reader (405 nm Absorbance). Absorbance outliers were excluded, and the triplicates averaged. The ELISA final titer was defined as the reciprocal of the final dilution at which the sample absorbance

was equal to or greater than twice the negative control absorbance. The negative control wells had all the components except the test sample.

## Plasma ADCC killing, ADCC titers, and V2-specific ADCC killing measured by F(ab')₂ blocking

ADCC killing, ADCC titers, and V2-specific ADCC assay were measured as previously described[27]. Cryopreserved plasma samples collected between 2 and 5 weeks following the last immunization (weeks 14–17) were analyzed. Briefly, $10^6$ GFP-expressing EGFP-CEM-NKr-CCR5-SNAP cells were coated with 50 µg of ΔV1gp120 protein for 2 h at 37 °C, washed and then labeled for 30 min at RT with SNAP-Surface® Alexa Fluor® 647 (New England Biolabs, Ipswich, MA). Total ADCC killing and ADCC titers were determined by incubating cells with plasma from each animal in the absence of F(ab')₂. In order to block the binding of plasma anti-V2 antibodies to the gp120, cells were incubated with 1 µg of purified F(ab')₂ fragments from NCI05 monoclonal antibody for 1 h at 37 °C. Plasma were heat-inactivated, diluted 1:100 for V2-specific ADCC or for 7 ten-fold dilutions (starting at 1:10) for total ADCC, and added to target cells. Target cells were then added of human PBMCs to generate an effector/target (E/T) ratio of 50:1. Cocultured cells were incubated at 37 °C for 2 h, washed with PBS and resuspended in 200 µl of a 2% paraformaldehyde in PBS solution. Cells were acquired by flow cytometry using an LSRII cytometer equipped with a high-throughput system (BD Biosciences). Specific killing was calculated as the loss of GFP from the SNAP-Alexa647⁺ target cells. Target and effector cells cultured in the presence of R10 media were used as background. Normalized percent killing was calculated as: (killing in the presence of plasma or plasma + F(ab')₂ – background)/(killing in the presence of positive control – background) × 100. Total ADCC killing was determined by incubating cells with plasma from each animal in the absence of F(ab')₂. The ADCC endpoint titer was calculated as the dilution at which the % ADCC killing was greater than the mean % killing of the background wells + three standard deviations. The V2-specific ADCC killing was calculated as: ADCC killing in the absence of F(ab')₂–ADCC killing in the presence of NCI05 F(ab')₂.

## Trogocytosis

Trogocytosis was evaluated as previously described[67] using cryopreserved plasma with plasma collected at baseline and 2- or 3-weeks following vaccination (weeks 14-15). CEM.NKR.CCR5 cells were stained with 2 µM PKH26 (Sigma-Aldrich) in Diluent C for 5 min at room temperature. Cells were then washed once, resuspended in R10 media, and incubated with wild-type gp120 or ΔV1gp120 proteins for 1 h at room temperature. Following gp120 coating, cells were washed, incubated with plasma samples diluted 300-folds. Healthy control PBMCs were used as effector cells. Cryopreserved PBMCS were thawed and cocultured with target cells (E:T ratio 50:1) for 5 h at 37 °C. At the end of the coculturing, cells were washed and incubated with live/dead aqua fixable dye and APC-H7-conjugated anti-CD14 antibody (5µl, clone MΦP9, Cat# 560180, BD Biosciences). Following incubation, cells were washed once and fixed with 4% formaldehyde (Tousimis, Rockville, MD). Cells were analyzed by flow cytometry using an LSRII flow cytometer (BD Biosciences). Trogocytosis score was calculated by calculating the PKH26 mean fluorescence intensity of the live CD14⁺ cells. Vaccine-induced trogocytosis scores were calculated by subtracting the values obtained for samples collected following the last immunization from those obtained for samples collected at baseline.

## Antibody-dependent neutrophil phagocytosis

ΔV1gp120 and wt gp120 proteins were biotinylated using EZ-Link Sulfo-NHS-LC-LC-Biotin and following the manufacturer's instructions (Cat. # 21338, Thermo Fisher Scientific), and using a biotin to gp120 ratio of 50. The biotin excess was then removed by using Zeba spin desalting columns (Cat. # PI-89883, Thermo Fisher Scientific) following the

manufacturer's instructions. Proteins were subsequently coupled with yellow-green or red NeutrAvidin-fluorescent beads (Cat. # F8776 and F8775, respectively; Life Technologies, Carlsbad, CA, USA) by incubating protein and beads at a 1:1 ratio for 2 h at 37 °C, washed and resuspended with 100x volume in 0.1% bovine serum albumin.

ADNP was assayed as already described[68]. Briefly, 10 µl of a 100-fold dilution of coupled protein/beads were incubated with 100 µl of 1:100 diluted plasma samples for 2 h at 37 °C. Following incubation, 50,000 cells/well of fresh peripheral blood leukocytes, isolated from one healthy donor, were added to the mix as effector cells, incubated for 1 h at 37 °C, washed, stained and fixed with 4% formaldehyde solution (Tousimis). Staining was conducted using Anti-human CD3 AF700 (5 µl, clone UCHT1, Cat# 557943) and anti-human CD14 APC-Cy7 (5 µl, clone MΦP9, Cat# 557831) antibodies obtained from BD Biosciences, and anti-human CD66b Pacific Blue (5 µl, clone G10F5, Cat# 305112) antibody from BioLegend. Cells were analyzed by flow cytometry using an LSRII flow cytometer (BD Biosciences). The phagocytic score was calculated by multiplying the % of bead-positive neutrophils (SSC high, CD3⁻ CD14⁻ CD66⁺) by the geometric MFI of the bead-positive cells and dividing by 104. Vaccine-induced ADNP was calculated by subtracting the values obtained for samples collected following the last immunization from those obtained for samples collected at baseline.

## Antibody-dependent cell phagocytosis

ΔV1gp120 and wt gp120 proteins were biotinylated and coupled with beads as described above for ADNP. ADCP was assayed as already described[69]. Briefly, 10 µl of a 100-fold dilution of coupled protein/beads were incubated with 100 µl of 1:100 diluted plasma samples for 2 h at 37 °C. Following incubation, 25,000 cells/well of THP-1 cells, were added to the mix as effector cells, incubated for 18 h at 37 °C, washed and fixed with 4% formaldehyde solution (Tousimis). Cells were analyzed by flow cytometry using an LSRII flow cytometer (BD Biosciences). The phagocytic score was calculated by multiplying the % of bead-positive cells by the geometric MFI of the bead-positive cells and dividing by 104. Vaccine-induced ADCP was calculated by subtracting the values obtained for samples collected following the last immunization from those obtained for samples collected at baseline.

## Efferocytosis

The frequency of CD14⁺ cells able to conduct efferocytosis and their capability to engulf apoptotic cells was determined by the Efferocytosis Assay kit (cat. #601770, Cayman Chemical Company, Ann Arbor, MI) as previously described[28]. Due to the lack of availability of cells collected from vaccinated animals of the alum group, the assay was conducted on $n = 6$ macaques of the alum group and $n = 12$ macaques of the ALFQA group.

**Effector cells.** Isolation of CD14⁺ cells was performed using non-human primate CD14 MicroBeads (#130-091-097) and AutoMACSpro (Miltenyi Biotec) starting from $10 \times 10^6$ cryopreserved PBMCs. Following isolation, cells were stained with CytoTell™ Blue provided in the kit and washed three times.

**Target cells.** Apoptotic neutrophils collected from a naïve macaque were used as target cells. The isolation of the neutrophils was performed by processing fresh blood collected in EDTA tubes by Ficoll Plaque (GE Healthcare) and subsequent incubation with Dextran. Following isolation, neutrophils were at first stained with CFSE, washed three times with R10 media (RPMI media, 10% FBS and 1X anti-anti), incubated with R10 media containing Staurosporine apoptosis-inducer, and finally washed two times.

**CD14⁺ cells and neutrophils coculture.** Effector and target cells were cocultured (ratio 1 effector to 3 target cells) in R10 media at a concentration of $1 \times 10^6$ cells/ml and incubated overnight in an incubator

at 37 °C. Concomitantly, CFSE-stained apoptotic neutrophils and CytoTell™ Blue-stained CD14$^+$ cells were cultured alone as controls. Following incubation, cells were washed once with PBS and resuspended in 1% paraformaldehyde. All cells were analyzed by flow cytometry, using a FACSymphony A5 and FACSDiva software (BD Biosciences). Further analyses were performed using FlowJo v10.1 (TreeStar, Inc.). Cells were gated as follows: FSC/SSC/Sigle cells/CytoTell™ Blue$^+$/CFSE$^+$. The % of CD14$^+$ efferocytes was calculated as the % of CFSE$^+$ cells in the CytoTell™ Blue$^+$ cells (CD14$^+$ cells), whereas the engulfing capability was determined as the median fluorescence intensity of CFSE in CFSE$^+$ CytoTell™ Blue$^+$ cells. The vaccine-induced % of CD14$^+$ efferocytes and their engulfing capability were calculated by subtracting the values obtained for samples collected 1 week following the last immunization (week 13) from those obtained for samples collected at baseline.

### Flow cytometry of rectal mucosal cells

The frequency of natural killer (NK)/Innate lymphoid cells (ILCs), macrophages and dendritic cells were measured in the rectal mucosa of macaques collected at baseline and 1 week following the last vaccination (week 13). Eleven freshly collected rectal biopsies were digested with collagenase (2 mg/ml; Sigma-Aldrich) in RPMI (Gibco) without FBS for 1 h at 37 °C. Following incubation, pinches were mechanically separated by using a 10 ml syringe with a blunt head canula, cells were washed with R10 and passed through a 70-μm cell strainer. Cells were counted and used for the experiment. For each sample, 10–15 million cells were recovered from the pinches.

**Phenotyping of innate lymphoid, myeloid and dendritic cells in rectal mucosa.** Two million cells were used for phenotype analysis. Cells were stained with Live/Dead blue dye (cat. #L34962, 0.5 μl) from Thermo Fisher, followed by surface staining for 30 min at RT with the following antibodies: BB700 anti-CD14 (M5E2; cat. # 745790, 5 μl), APC anti-CCR2 (48607; cat. # 558406, 5 μl), APC-Cy7 anti-HLA-DR (L243; cat. # 335796, 5 μl), BV480 anti-CD45 (D058-1283; cat. # 566145, 5 μl), BV650 anti-NKp44 (p44-8; cat. # 744302, 5 μl), BV750 anti-CD163 (GHI/61; cat. # 747185, 5 μl), BUV493 anti-CD73 (AD2; cat. # 750061, 5 μl), BUV563 anti-CD184(CXCR4) (12G5; cat. # 741400, 5 μl), BUV661 anti-CD141 (1A4; cat. # 741650, 5 μl), BUV737 anti-CD206 (19.2; cat. # 741860, 5 μl), BUV805 anti-CD3 (SP34-2; cat. # 742053, 5 μl), BUV805 anti-CD20 (2H7; cat. # 612905, 5 μl) from BD Biosciences (San Jose, California, USA); PE-Cy7 anti-NKG2A (Z199; cat. no. B10246, 5 μl) from Beckman Coulter (Brea, California, USA); AF488 anti-CD1a (O10; cat. No. NBP2-34697AF488, 5 μl) from Novus Biologicals (Centennial, Colorado, USA); PE anti-CD33 (AC104.3E3; cat. #130-113-349, 5 μl) from Miltenyi Biotec (Bergisch Gladbach, North Rhine-Westphalia, Germany); and PE/Dazzle594 anti-CD16 (3G8; cat. # 302054, 5 μl), PE-Cy5 anti-CD11c (3.9; cat. # 301610, 5 μl), BV570 anti-CD11b (ICRF44; cat. # 301325, 5 μl), BV605 anti-CD1c (L161; cat. # 331538, 5 μl) from BioLegend (San Diego, California, USA). Samples were acquired on a BD FACSymphony A5 cytometer and analyzed with FlowJo software 10.6.

NKG2A$^+$ NK cells were gated as singlets/live/CD45$^+$/CD3$^-$CD20$^-$/CD14$^-$/NKG2A$^+$NKp44$^-$ cells. NKp44$^+$ cells were gated as singlets/live/CD45$^+$/CD3$^-$CD20$^-$/CD14$^-$/NKG2A$^-$NKp44$^+$ cells. NKG2A$^-$ NKp44$^-$ cells were gated as singlets/live/CD45$^+$/CD3$^-$/CD20$^-$/CD14$^-$/NKG2A$^-$NKp44$^-$ cells. CD73$^+$ macrophages were gated as singlets/live/CD45$^+$/CD3$^-$CD20$^-$/CD11b$^+$/HLA-DR$^+$/FSC-A$^{High}$SSC-A$^{High}$/CD163$^+$/CD73$^+$ cells and expressed as frequency of parental population. Dendritic DC-10 cells were gated as previously described[28] as singlets/SSC$^{high}$FSC$^{high}$/live/CD45$^+$/CD3$^-$CD20$^-$/HLA-DR$^+$/CD1c$^-$/CD11b$^+$/CD11c$^+$/CD14$^-$CD16$^+$/CD163$^+$/CD141$^+$/CD1a$^-$ and expressed as frequency of CD45$^+$ cells.

Since the protocol allowed for the collection of a maximum of 11 rectal mucosa biopsies, which usually yield between 10 and 15 million cells, it was not possible to perform concomitant gate validation. Therefore, validation of the gates for NKG2A, Nkp44, CD163, CD141,

and CD73 was performed at a later time point and using mucosal samples collected from two animals enrolled in another study. Briefly, following cell isolation, cells were divided into six tubes. One tube was fully stained following the procedure described above, whereas the other five were stained with all the antibodies minus one, constituting the fluorescence minus one (FMO) tubes of NKG2A, Nkp44, CD163, CD141 and CD73 antibodies. Examples of the gating in full-stained and FMO tubes are reported in Supplementary Fig. 3b. Further validation of the gate positioning in the original dataset was performed by gating each marker on live CD45$^+$ cells. Briefly, using the same FCS files used to generate the data reported here, each marker was gated in the total live CD45$^+$ population (Supplementary Fig. 3c). The presence of a high number of positive and negative cells in the CD45$^+$ population allowed an accurate positioning of the gates. The generated gates were then applied in the full gating strategies (Supplementary Fig. 3a). Examples of the gating on live CD45$^+$ cells are reported in Supplementary Fig. 3c.

Vaccine-induced frequencies of cells were calculated by subtracting the values obtained for samples collected following last immunization of those obtained for samples collected at baseline.

**Cytokine expression upon stimulation of innate lymphoid cells in rectal mucosa.** Two million rectal mucosal cells were stimulated for 2 h at 37 °C with overlapping gp120 peptides encompassing the sequence of SIV$_{mac251}$ gp120 or HIV-1 A244 gp120 (2 μg/ml), or 1X PMA/Ionomycin (eBioscience cell stimulation cocktail, Cat. #00-4970-93 Invitrogen). Subsequently, GolgiPlug protein transport inhibitor (containing Brefeldin A) (cat. #555029, 1 μl) and GolgiStop protein transport inhibitor (containing Monensin) (cat. #554724, 0.7 μl) were added and culturing continued for 18 hours. Following incubation, cells were stained with Live/Dead blue dye (cat. #L34962, 0.5 μl) from Thermo Fisher, followed by surface staining for 30 min at room temperature. Surface staining was conducted as described above for phenotyping of innate lymphoid, myeloid and dendritic cells in rectal mucosa. Following surface staining, cells were fixed and permeabilized with a FOXP3-transcription buffer set (cat. #00-5523-00) from eBioscience (San Diego, California, USA) according to the manufacturer's recommendation and subsequently intracellular staining with the following: R718 anti-TNFα (MAb11; cat. #566957, 5 μl), BV711 anti-IL-10 (JES3-9D7; cat. # 564050, 5 μl), BV786 anti-CD107 (H4A3; cat. # 563869, 5 μl), BUV395 anti-IFN-γ (B27; cat. # 563563, 5 μl) from BD Biosciences (San Jose, California, USA); and BV421 anti-IL-17 (BL168; cat. # 512312, 5 μl) from BioLegend (San Diego, California, USA). Samples were acquired on a BD FACSymphony A5 cytometer and analyzed with FlowJo software 10.6. Cytokines were gated on the parent population in NKG2A$^+$ NK cells, NKp44$^+$ and NKG2A$^-$ NKp44$^-$ cells.

### Proximity extension assay

Protein quantification was executed employing the Olink® Target 48 Cytokine panel* (Olink Proteomics AB, Uppsala, Sweden) in accordance with the manufacturer's protocols. Cryopreserved plasma collected from naïve control animals, and at 24 h or 1 week following the last immunization were analyzed. This method leverages the proximity extension assay (PEA) technology as detailed extensively by ref. [70]. This specific PEA methodology enables the concurrent assessment of 45 distinct analytes. Briefly, pairs of oligonucleotide-labeled antibody probes, each tailored to selectively bind to their designated protein targets, were used. Probe pairs mix were incubated with 1 μl of plasma. Probes that encountered their cognate proteins are then in close spatial proximity, and their respective oligonucleotides engage in pairwise hybridization. A DNA polymerase was used to amplify the hybridized DNA duplex, and to create distinct PCR target sequences. Subsequently these newly formed DNA sequences were detected and quantified through utilization of a microfluidic real-time PCR platform, specifically the Biomark HD system by Fluidigm (Olink Signature Q100 instrument). Data validation to uphold data integrity was conducted

with the Olink NPX Signature software, specifically designed for the Olink® analysis: the application was used to import data from the Olink Signature Q100 instrument and process the data. Data normalization procedures were executed employing an internal extension control and calibrators, thereby effectively mitigating any inherent intra-run variability. The ultimate assay output was reported in picograms per milliliter (pg/ml), predicated upon a robust 4-parameter logistic (4-Pl) fit model, thereby ensuring precise absolute quantification. Comprehensive insights into the assay's validation parameters, encompassing limits of detection, intra- and inter-assay precision data, and related metrics, are available at www.olink.com.

Output from the Olink software was further processed to extrapolate values for samples that were below the lower limit of quantification (LLOQ) and above the upper limit of quantification (annotated as ">ULOQ"). The Olink software interpolates values below the LLOQ through fitting the 4-Pl model to a distinct minimum limit of detection for each plate of samples run, and values below this interpolation range are set to NaN. Since these values are below the limit of detection but not truly missing, for each assay, we determined a universal "below detection" value by taking the mode LLOQ across 22 plates, divided by 10,000 (which was below all interpolated values in this extensive historical dataset) and set all NaN to this assay-specific below detection value. Samples which were above the ULOQ were set to the ULOQ value for the indicated target assay from the plate on which the sample was run. Finally, true missing values annotated as "No Data" were converted to NA to be systematically treated as missing. For Fig. 4c, d, the values plotted on the x-axis were calculated as the $\log_{10}(abs(MW$ estimate *100)) * sign of MW estimate. The MW estimate = median of outer differences, and outer differences = difference between all pairs of values in group Alum−group ALFQA (e.g., for A = a1, a2 and B = b1, b2; then the outer differences would be: a1-b1, a2-b1, a1-b2, a2-b2. And the MW estimate is the median of these deltas). For MW estimate >0, ALFQA > ALUM; and for MW estimate <0, ALFQA < ALUM. The *100 of the formula is applied to shift all estimates >1 (o $R < -1$) so that the $\log_{10}$ does not transform values $-1 < x < 1$ [thus for a MW estimate of 0.01 (the smallest magnitude estimate) -> $\log_{10}(0.01*100) = 0$].

For Supplementary Fig. 5d, e, mean $\log_{10}$ values for each experimental group were visualized relative to the distribution of historical values for each assay (grey boxplot, with grey triangle designating the mean). Mann−Whitney/Wilcoxon $p < 0.05$ between ALUM and ALFQA groups is designated by a star. Assays were sorted on the y-axis by MW result category and then by the mean of the historical baseline. Plots were generated using R version 4.4.1. In particular, ggalluvial_0.12.5 was used for alluvial diagrams, corrplot_0.95 was used for correlation heatmaps, and ggpubr_0.6.0 was used for the other plots.

### Proteome analysis by ingenuity pathway analysis
Qiagen ingenuity pathway analysis application (Version 01-23-01; Qiagen Sciences, Germantown, MD, USA) was used to perfume pathway analysis. Data for analysis was uploaded as fold change of the mean of the ALFQA group compared to the mean of the alum group for each detectable target of the proteome (Fold change = mean ALFQA/mean alum). Each fold change was uploaded on the application in association with the $p$ values and adjusted $p$ values (false discovery rate, Benjamini−Hochberg) of the comparison of mean ALFQA vs mean alum. Data obtained from samples collected at week 12 + 24 h and week 13 were analyzed independently. Only the 36 detectable targets were included in the analysis. For the graphical summary (Fig. 5b, c), nodes were kept unmodified. For pathways (Supplementary Fig. 6a, b), only pathways with significance higher than −log($p$ value) 10 were exported.

### Statistical analysis
GraphPad Prism (Version 10.3.0 (461)) was used to calculate the statistics. The Mann−Whitney−Wilcoxon test was used to perform comparisons of continuous factors between macaques belonging to the alum and ALFQA groups. All scatter plots report the median, except where stated otherwise. Spearman's rank correlation was used to infer linear relationships between measured variables. The number of challenges is a numeric variable taking integer values between 1 and 11 with right-censoring of higher values, recorded as >11. For graphing, we assigned 12 as the value of the right-censored numbers, but the ranks are the same for any assigned value greater than 11. The exact log-rank Mantel−Cox test of the discrete-time proportional hazards model was used to compare the viral acquisition in animals exposed to $SIV_{mac251}$. All statistical tests were performed as two-tailed.

### Reporting summary
Further information on research design is available in the Nature Portfolio Reporting Summary linked to this article.

## Data availability
Source Data are provided with this manuscript in the Source Data file. The raw data and files used to generate the findings presented in this manuscript have been deposited in the Zenodo repository at https://zenodo.org/records/15602857 and can be cited using https://doi.org/10.5281/zenodo.15602857.

## Code availability
The source codes for the analyses are available at: https://github.com/NCI-VB/franchini_bissa_ALFQAvsAlum, and can be cited using https://doi.org/10.5281/zenodo.15747628.

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

## Acknowledgements

We thank David Ahern for editorial assistance. We thank Nancy Miller and John Warren (NIAID) for discussion, James Tartaglia (Sanofi Pasteur) for the ALVAC-SIV vaccine, and George N. Pavlakis, Barbara K. Felber, and Margherita Rosati (NCI) for the SIV gag DNA vaccine used in the study. We thank Camille M. Lange for assistance with the ALFQA. This research was supported in part by the Intramural Research Program of the National Institutes of Health (NIH). The contributions of the NIH author(s) were made as part of their official duties as NIH federal employees, are in compliance with agency policy requirements, and are considered Works of the United States Government. However, the findings and conclusions presented in this paper are those of the author(s) and do not necessarily reflect the views of the NIH or the U.S. Department of Health and Human Services. This work was supported with federal funds from the National Cancer Institute Intramural Program and the Office of AIDS Research (OAR), National Institutes of Health (to G. Franchini). The contents of this publication do not necessarily reflect the views or policies of the Department of Health and Human Services, nor does mention of trade names, commercial products, or organizations imply endorsement. Material has been reviewed by the Walter Reed Army Institute of Research. There is no objection to its presentation and/or publication. The opinions or assertions contained herein are the private views of the author, and are not to be construed as official, or as reflecting true views of the Department of the Army or the Department of Defense. This work was supported by agreements #W81XWH-18-2-0040 and #HT94252430004 between the Henry M. Jackson Foundation for the Advancement of Military Medicine, Inc., and the US Department of Defense (DOD). SIVmac251 virus supplied by QBI and the Vaccine Research Program, Division of AIDS, NIAID under Contract No. 75N93025C00011. Plasma neutralizing antibodies analysis was supported by NIAID Contract #75N93025C00006.

## Author contributions

M.B. and G.F. conceived the studies and wrote the paper, with contributions from all authors; M.B. coordinated and performed the macaque studies with M.A.R., L.S., E.K.W., S.B., M.N.D., R.W.-P., and S.S.; I.S.d.C. and A.G. performed antibody titers assays; J.K., S.B., and R.A. performed mucosal antibody titers and antibody avidity assays; M.A.R. performed ADCC and V2-ADCC assays; M.B., A.G., and N.R.K. performed the efferocytosis assay; D.P.-P. and K.F.N. performed trogocytosis, ADCP, and ADNP; M.A.R., A.G., K.M., and S.B. performed flow cytometry assays; L.S. performed proximity extension assay; G.R.M. and M.R. provided the ALFQ; L.S. and K.C.G. performed analysis of proximity extension assay data; K.C.G. performed computational analyses; X.S. and D.C.M. performed neutralization antibody assay; C.A.P.-M. and T.C. reviewed the manuscript and provided intellectual insights.

## Funding

## Competing interests
The authors declare no competing interests.

## Additional information

Massimiliano Bissa [1,11] ✉, Mohammad Arif Rahman [1,11], Luca Schifanella[1,11], Katherine C. Goldfarbmuren [2,3,11], Isabela Silva de Castro[1,11], Emmanuel K. Woode[1], Anna Gutowska [1], Melvin N. Doster[1], Sophia Brown[1,4], Sarkis Sarkis [1], Neil R. Kanchetty[1], Cynthia A. Pise-Masison[1], Robyn Washington-Parks[1], Katherine McKinnon[4], Shraddha Basu[5,6], Jiae Kim [5,6], Ryan Alving[7], Dominic Paquin-Proulx [5,6], Kombo F. N'guessan[5,6], Xiaoying Shen [8,9], David C. Montefiori [8,9], Timothy Cardozo [10], Gary R. Matyas [5], Mangala Rao[5] & Genoveffa Franchini [1] ✉

[1]Animal Models and Retroviral Vaccines Section, National Cancer Institute, Bethesda, MD, USA. [2]Advanced Biomedical Computational Science, Frederick National Laboratory for Cancer Research, Frederick, MD, USA. [3]Center for Cancer Research Collaborative Bioinformatics Resource, National Cancer Institute, Bethesda, MD, USA. [4]CCR Building 41 Flow Cytometry Core, National Cancer Institute, Bethesda, MD, USA. [5]U.S. Military HIV Research Program, Center for Infectious Disease Research, Walter Reed Army Institute of Research, Silver Spring, MD, USA. [6]U.S. Military HIV Research Program, Henry M. Jackson Foundation for the Advancement of Military Medicine, Bethesda, MD, USA. [7] Oak Ridge Institute of Science and Education, Oak Ridge, TN, USA. [8]Department of Surgery, Duke University, Durham, NC, USA. [9]Duke Human Vaccine Institute, Duke University, Durham, NC, USA. [10]New York University Grossman School of Medicine, NYU Langone Health, New York, NY, USA. [11]These authors contributed equally: Massimiliano Bissa, Mohammad Arif Rahman, Luca Schifanella, Katherine C. Goldfarbmuren, Isabela Silva de Castro. ✉e-mail: massimiliano.bissa@nih.gov; franchig@mail.nih.gov

