## [Transparent Peer Review file · Nature Communications]

HIV vaccine candidate mucosal immunity in female macaques augmented by $\Delta V1gp120$ formulated in ALFQA adjuvant

Corresponding Author: Dr Genoveffa Franchini

Version 0:

Reviewer comments:

Reviewer #1

(Remarks to the Author)

The authors compared SIV- and HIV-derived VLPs coated with V1-deleted Env, administered via DNA/ALVAC followed by a protein $\Delta V1/gp120$ boost, with the protein formulated using ALFQ alone or with alum as AFLQA. The former achieved 60% protection for challenged macaques, while the latter enhanced protection to 79%. Several thorough analyses were conducted to understand the protection mechanisms, including antibody levels, ADCC, efferocytosis, and mucosal cell characterisation. Plasma proteomics highlighted the concentration of CCL8.

The work is well conducted, the manuscript is well written, and presents a large amount of data that will be of interest to specialised readers in the field.

Reviewer #2

(Remarks to the Author)

This research paper by Massimiliano Bissa et. al. investigates the efficacy of an SIV vaccine that utilizes "ALFQA", an MPLA/saponin/alum adjuvant instead of Alum alone in the final boost. 12 animals were vaccinated and immune responses elicited in the mucosa and plasma were monitored as was viremia after mucosal challenge with SIVmac251. Results were compared with previous experiments in 37 unvaccinated animals or in 30 animals where the Alum adjuvant was used in the final boost. These animals were challenged with the same stock and dose.

Key findings of the study are: 1) ALFQA increases anti-inflammatory M2-like macrophages, DC-10 dendritic cells and NKp44+ innate lymphoid cells (ILCs) at the mucosa. 2) V2-specific ADCC and DC-10 frequencies correlate with a decreased risk of SIVmac251 infection. This is consistent with non-neutralizing V2-antibody titers correlating with protection in the RV144 clinical trial. 3) Plasma lymphotoxin-alpha (TNF-beta) and CCL8 levels associated with vaccine-induced protection. 4) ALFQA improves Fc-mediated effector functions, including ADCC, ADNP and efferocytosis. These findings were all statistically significant, however adjuvant mediated improvements in protection compared to Alum vaccinated animals was not achieved. The work directly informs the upcoming CLEAR Phase I clinical trial (2025) and contributes to improving HIV vaccine efficacy.

Strengths of this paper include thorough evaluation of plasma and mucosal immune responses leading to the identification of statistical differences in specific mucosal immune parameters for Alum vs. ALFQA adjuvanted vaccines. This will offer significant translational relevance for similarly designed clinical trials. While the study reports a 79% reduction in infection here, the protection attributed to ALFQA adjuvant was not statistically significant (ALUM offered 59%). This should be explicitly stated in the paper in addition to providing the poor p-value in figure 1D. While autologous neutralizing antibodies to SIVmac251 are not necessarily expected by this vaccine regimen, they should at least be evaluated for publication since their role in protection could outsize fc-mediated correlates. Finally, while the authors offer a reasonable hypothesis for why ALFQA adjuvanted vaccines may offer improved protection against mucosal SIV challenge; i.e. reducing inflammation and therefore viral target CD4-T cells at infection sites, the durability of such effects are not investigated and it seems likely that differences in innate immune cell frequencies might dissipate within 2-3 months. Durability of vaccine induced immune changes at the mucosa should at least be considered in the discussion.

(Remarks to the Author)

Bissa et al's study presents valuable findings regarding the enhancement of SIV protection through a novel vaccine formulation. The authors investigated the impact of ALFQ in combination with $\Delta V1gp120$ formulated in Alum (ALFQA) on protection against SIVmac251 acquisition. Their findings indicate that boosting with $\Delta V1gp120$ in ALFQA resulted in a 79% reduction in the risk of infection, compared to a 59.8% reduction observed in animals vaccinated with $\Delta V1gp120$ formulated in Alum alone. The protective effect was linked to enhanced antibody responses to V2, increased V2-specific ADCC activity, CD14+ cell-mediated efferocytosis, and the induction of a mucosal tolerogenic immune response.

This result is of interest as it demonstrates an enhanced, although moderate efficacy of the ALFQA formulation, indicating that the addition of ALFQ to $\Delta V1gp120$ may provide a more robust immune response.

The manuscript is well-written, and the methodology is, overall, well-explained. However, additional details in the methods section, particularly regarding flow cytometry analysis, would strengthen the paper. The manuscript could be further improved as outlined below:

- 1) Abstract and Results (Line 126): There is a discrepancy between the abstract and results section regarding vaccine efficacy. The abstract reports 60% efficacy for the ALFQA group, whereas the results section states 79%. To ensure consistency and clarity, the authors should align the reported efficacy percentages. If the 79% efficacy in the results section is correct, the abstract should be revised accordingly. Furthermore, the abstract should clarify whether ALFQA confers significantly greater protection than Alum or if the difference remains a statistical trend.
- 2) Results (Line 129): The wording is ambiguous regarding the vaccination of the two groups of macaques. As written, it suggests that all animals were newly immunized. However, since the Alum-only group (n=30) has been previously published, it is unclear whether these animals were part of a new experiment or if their data were reanalyzed. If the latter is true, the authors should explicitly state that the Alum-only data were derived from prior studies, while the ALFQA group represents a new experimental cohort. Clarity can be improved by stating, for example, "Data from the Alum-only group were previously published and are included here for comparison with the newly vaccinated ALFQA group." Conversely, if all animals were newly vaccinated, the text should be revised to remove ambiguity.
- 3) Results (Line 142): To improve clarity, the authors should specify the exact number of infected animals in each group rather than requiring the reader to count them in Supplementary Fig. 1a-g. This would facilitate data interpretation and better assessment of vaccine efficacy.
- 4) Supplementary Fig. 3: The gating strategy used to identify CD73+ CD163+ macrophages in rectal mucosa appears to include many events along the axes in the dot plots. The authors should provide appropriate fluorescence-minus-one (FMO) controls to assess the accuracy of gating.
- 5) How were DC10 identified in rectal tissues? A representative flow plot showing the gating strategy with appropriate FMO controls should be included. The same applies to the identification of ILCs.
- 6) The induction of CD16+CD73+ macrophages and DC10 is interesting, but it is unclear whether these cell populations are more efficiently induced in ALFQA-vaccinated macaques compared to Alum, which would explain the higher vaccine efficacy observed with ALFQA. This is particularly relevant given that CD73+ M2-like macrophages and CD73+ DC-10 were positively correlated in both Alum and ALFQA groups (Supplementary Fig. 4f) and that DC10 frequencies did not significantly differ between the two vaccine groups.
- 7) The authors emphasize DC10 induction as a major protective mechanism in ALFQA vaccination due to the correlation between DC10 frequency and lymphotoxin- α . However, no significant induction of DC10 is shown. This should be addressed.
- 8) Line 246: was the correlation between DC10 frequency and the number of challenges required for protection in ALFQA also observed in Alum group? Clarifying this point would strengthen the interpretation of the data.
- 9) The finding that ALFQA vaccination had an opposite effect on CD73 expression in DC10 versus M2-like macrophages, leading to lower CD73+ DC10 compared to Alum (p=0.024; Supplementary Fig. 4e) is intriguing and deserves further discussion.
- 0) Second study with HIV immunogens: It is unclear whether mucosal samples were collected at baseline, as was done in the first study (line 268 vs. line 285). If baseline values were not obtained, it becomes difficult to determine whether CD73+CD163+ macrophages (Fig. 3d) were significantly induced or if higher levels were already present at baseline. Additionally, the first study presents results as fold change from baseline (Fig. 3a), making it impossible to compare the two datasets directly. If baseline data are missing, can the authors provide data from the control group for comparison? Furthermore, Fig. 3a appears to show absolute frequencies rather than fold changes. The authors should verify whether the y-axis legend in Fig. 3a is correctly labeled.
- 1) Did the authors assess IL-10 production by DC10? Additionally, were type 1 regulatory CD4 T cells (Tr1) induced by DC10 in this experimental setting? Addressing these questions would provide further mechanistic insights.

Minor comments

Results (Lines 151, 159): The references to supplementary figures should be corrected: Supplementary Fig. 1a should be 2a, and Supplementary Fig. 1b-c should be 2b-c.

Line 179: Please specify that the analysis was performed 2–3 weeks following the last vaccination.

Version 1:

Reviewer comments:

Reviewer #2

(Remarks to the Author)

The authors have addressed reviewer concerns to improve the quality and clarity of the work. The new statements in the paper should be reviewed and modified for improved style/grammar. For example line 149 should read "the ALFQA and Alum groups..." and line 521 should read "Here, we identified the important role of..." etc.

Reviewer #3

(Remarks to the Author)

The authors have thoroughly addressed all the points raised by this reviewer in the initial round. I am satisfied with the revisions and appreciate the effort invested in improving the manuscript. Thank you, and congratulations on your work.

Reviewer #1 (Remarks to the Author):

The authors compared SIV- and HIV-derived VLPs coated with V1-deleted Env, administered via DNA/ALVAC followed by a protein deltaV1/gp120 boost, with the protein formulated using ALFQ alone or with alum as AFLQA. The former achieved 60% protection for challenged macaques, while the latter enhanced protection to 79%. Several thorough analyses were conducted to understand the protection mechanisms, including antibody levels, ADCC, efferocytosis, and mucosal cell characterization. Plasma proteomics highlighted the concentration of CCL8.

The work is well conducted, the manuscript is well written, and presents a large amount of data that will be of interest to specialized readers in the field.

We thank reviewer #1 for appreciating our work and providing a positive

evaluation. Reviewer #2 (Remarks to the Author):

This research paper by Massimiliano Bissa et. al. investigates the efficacy of an SIV vaccine that utilizes “ALFQA”, an MPLA/saponin/alum adjuvant instead of Alum alone in the final boost. 12 animals were vaccinated and immune responses elicited in the mucosa and plasma were monitored as was viremia after mucosal challenge with SIVmac251. Results were compared with previous experiments in 37 unvaccinated animals or in 30 animals where the Alum adjuvant was used in the final boost. These animals were challenged with the same stock and dose.

Key findings of the study are: 1) ALFQA increases anti-inflammatory M2-like macrophages, DC-10 dendritic cells and NKp44+ innate lymphoid cells (ILCs) at the mucosa. 2) V2-specific ADCC and DC-10 frequencies correlate with a decreased risk of SIVmac251 infection.

This is consistent with non-neutralizing V2-antibody titers correlating with protection in the RV144 clinical trial. 3) Plasma lymphotoxin-alpha (TNF-beta) and CCL8 levels associated with vaccine-induced protection. 4) ALFQA improves Fc-mediated effector functions, including ADCC, ADNP and efferocytosis. These findings were all statistically significant, however adjuvant mediated improvements in protection compared to Alum vaccinated animals was not achieved. The work directly informs the upcoming CLEAR Phase I clinical trial (2025) and contributes to improving HIV vaccine efficacy.

Strengths of this paper include thorough evaluation of plasma and mucosal immune responses leading to the identification of statistical differences in specific mucosal immune parameters for Alum vs. ALFQA adjuvanted vaccines. This will offer significant translational relevance for similarly designed clinical trials.

1) While the study reports a 79% reduction in infection here, the protection attributed to ALFQA adjuvant was not statistically significant (ALUM offered 59%). This should be explicitly stated in the paper in addition to providing the poor p-value in figure 1D.

We agree with the reviewer #2 and we have underlined that viral acquisition did not differ significantly between the two vaccinated groups, when compared against each other (lines 144-147)

2) While autologous neutralizing antibodies to SIV_{mac251} are not necessarily expected by this vaccine regimen, they should at least be evaluated for publication since their role in protection could outsize fc-mediated correlates.

We agree with the reviewer #2 of the importance of evaluating the ability of the 2 regimens in eliciting neutralizing antibodies. We therefore measured the levels of neutralizing antibodies, and we have incorporated them in the results section as well in the methods (lines 168-174, 814-828 and **Supplementary table 3**). As reported, and surprisingly the neutralizing antibody titers against SIV strains were lower in ALFQA treated animals and as expected with this vaccine modality their level did not correlate with decreased virus acquisition.

3) Finally, while the authors offer a reasonable hypothesis for why ALFQA adjuvanted vaccines may offer improved protection against mucosal SIV challenge, i.e. reducing inflammation and therefore viral target CD4-T cells at infection sites, the durability of such effects are not investigated, and it seems likely that differences in innate immune cell frequencies might dissipate within 2-3 months. Durability of vaccine induced immune changes at the mucosa should at least be considered in the discussion.

We have included in the discussion the possible limit of innate immune response durability and stated that further studies will be needed to assess long-term durability of adaptive and innate vaccine-induced immunity (lines 521-527).

Reviewer #3 (Remarks to the Author):

Bissa et al's study presents valuable findings regarding the enhancement of SIV protection through a novel vaccine formulation. The authors investigated the impact of ALFQ in combination with AV1gp120 formulated in Alum (ALFQA) on protection against SIV_{mac251} acquisition. Their findings indicate that boosting with AV1gp120 in ALFQA resulted in a 79% reduction in the risk of infection, compared to a 59.8% reduction observed in animals vaccinated with AV1gp120 formulated in Alum alone. The protective effect was linked to enhanced antibody responses to V2, increased V2-specific ADCC activity, CD14+ cell-mediated efferocytosis, and the induction of a mucosal tolerogenic immune response. This result is of interest as it demonstrates an enhanced, although moderate efficacy of the ALFQA formulation, indicating that the addition of ALFQ to AV1gp120 may provide a more robust immune response. The manuscript is well-written, and the methodology is, overall, well-explained. However, additional details in the methods section, particularly regarding flow cytometry analysis, would strengthen the paper. The manuscript could be further improved as outlined below:

1) Abstract and Results (Line 126): There is a discrepancy between the abstract and results section regarding vaccine efficacy. The abstract reports 60% efficacy for the ALFQA group, whereas the results section states 79%. To ensure consistency and clarity, the authors should align the reported efficacy percentages. If the 79% efficacy in the results section is correct, the

abstract should be revised accordingly. Furthermore, the abstract should clarify whether ALFQA confers significantly greater protection than Alum or if the difference remains a statistical trend.

We agree with the reviewer #3 that this may be confusing for the readers. We have therefore specified the vaccine efficacy in the abstract (line 33), in addition to the already acknowledged % of uninfected animals. In addition, in the results section, we have added the % of uninfected (protected from infection) animals following termination of challenge exposure (lines 147-149)

2) Results (Line 129): The wording is ambiguous regarding the vaccination of the two groups of macaques. As written, it suggests that all animals were newly immunized. However, since the Alum-only group (n=30) has been previously published, it is unclear whether these animals were part of a new experiment or if their data were reanalyzed. If the latter is true, the authors should explicitly state that the Alum-only data were derived from prior studies, while the ALFQA group represents a new experimental cohort. Clarity can be improved by stating, for example, “Data from the Alum-only group were previously published and are included here for comparison with the newly vaccinated ALFQA group.” Conversely, if all animals were newly vaccinated, the text should be revised to remove ambiguity.

We further clarified this point by revising the first paragraph of results as suggested by the reviewer (lines 131-135).

3) Results (Line 142): To improve clarity, the authors should specify the exact number of infected animals in each group rather than requiring the reader to count them in Supplementary Fig. 1a-g. This would facilitate data interpretation and better assessment of vaccine efficacy.

As suggested, we have included the numbers of protected from infection or infected animals in each group in the results section (lines 148)

4) Supplementary Fig. 3: The gating strategy used to identify CD73⁺ CD163⁺ macrophages in rectal mucosa appears to include many events along the axes in the dot plots. The authors should provide appropriate fluorescence-minus-one (FMO) controls to assess the accuracy of gating.

The gating strategy has been updated, as well as the material and methods section (lines 1007-1021). In particular, we have explained and showed how we determined the position of the gates for identifying macrophages, dendritic and innate lymphoid cells (lines 1007-1021; **Supplementary figure 3a and b**), in mucosal samples.

5) How were DC10 identified in rectal tissues? A representative flow plot showing the gating strategy with appropriate FMO controls should be included. The same applies to the identification of ILCs.

Representative flow for CD73⁺CD163⁺ macrophages, DC-10 and ILCs were included in the supplementary figures (**Supplementary Fig 3a and b**), as requested.

6) The induction of CD163⁺CD73⁺ macrophages and DC10 is interesting, but it is unclear whether these cell populations are more efficiently induced in ALFQA-vaccinated macaques

compared to Alum, which would explain the higher vaccine efficacy observed with ALFQA. This is particularly relevant given that CD73⁺ M2-like macrophages and CD73⁺ DC-10 were positively correlated in both Alum and ALFQA groups (Supplementary Fig. 4f) and that DC10 frequencies did not significantly differ between the two vaccine groups.

The data reported in the Figure 3a demonstrate that effect of the two adjuvants in the induction of CD73⁺CD163⁺ macrophages is significantly different, showing that ALFQA favors more CD73⁺CD163⁺ macrophages than Alum. On the other hand, percentage of vaccine-induced DC-10 did not differ with the two regimens, suggesting that their function might be different (perhaps higher production of IL-10?), as the levels of DC-10 correlate with reduced risk of SIy acquisition. We have added this point to the results and clarified that further studies are needed to evaluate the functionality and role of these cells (lines 263-267)

7)The authors emphasize DC10 induction as a major protective mechanism in ALFQA vaccination due to the correlation between DC10 frequency and lymphotoxin- α . However, no significant induction of DC10 is shown. This should be addressed.

We agree with the reviewer. We believe that LTA may influence the function of the DC-10, and not their frequency. However, due to the lack of fresh mucosal samples from vaccinated animals, this hypothesis cannot be tested in the current study. The speculation about the possible effect of the LTA on the function of DC-10 has been included in the discussion and clarified that additional studies on the function of DC-10 will be required (lines 508-520).

8)Line 246: was the correlation between DC10 frequency and the number of challenges required for protection in ALFQA also observed in Alum group? Clarifying this point would strengthen the interpretation of the data.

As specified in the method section (lines 728-738), the animals in Alum group used for the analysis of the mucosal immune responses were part of another study (where a microbicide was added following mucosal samples collection and during challenge exposure), therefore the challenge results could not be compared (**Supplementary Fig. 4a**) This hypothesis need to be tested in an additional experiment. This has been clarified in the results (lines 263-267)

9)The finding that ALFQA vaccination had an opposite effect on CD73 expression in DC10 versus M2-like macrophages, leading to lower CD73⁺ DC10 compared to Alum (p=0.024;Supplementary Fig. 4e) is intriguing and deserves further discussion.

Extensive literature is available regarding the expression of CD73 on macrophages, on the other hand CD73 expression in DC-10 has never been reported. In skin dendritic cells the expression of CD73 affects their migration. Although we do not know whether this is effect or others may affect the frequency of these cells, in the manuscript we have discussed this role of CD73 in other dendritic cells and added that additional studies on the expression of CD73 in DC-10 are required (lines 513-520).

0) Second study with HIV immunogens: It is unclear whether mucosal samples were collected at baseline, as was done in the first study (line 268 vs. line 285). If baseline values were not obtained, it becomes difficult to determine whether CD73+CD163+ macrophages (Fig. 3d) were significantly induced or if higher levels were already present at baseline. Additionally, the first study presents results as fold change from baseline (Fig. 3a), making it impossible to compare the two datasets directly. If baseline data are missing, can the authors provide data from the control group for comparison? Furthermore, Fig. 3a appears to show absolute frequencies rather than fold changes. The authors should verify whether the y-axis legend in Fig. 3a is correctly labeled.

Mucosal biopsies in the HIV study were collected only following vaccination. We have specified it in the results (line 287). Additionally, the study did not include control animals. We understand the concerns of the reviewer, however in that study the animals were properly randomized, and all the vaccination and collections were performed concomitantly decreasing the chance of a differential baseline effect. We explain this point in the result session (lines 742-757). In the SIV study, frequencies of CD73+CD163+ macrophages at baseline ranged between 10.81-18.63% for ALFQA and 5.55-21.05% for Alum, whereas frequencies at post vaccination ranged between 54.25-85.15% for ALFQA and 41.42-68.63% for Alum. Data reported in Fig.3a are correct and represent the vaccine-induced variation of the frequencies, expressed as the subtraction of % at post vaccination - % at baseline.

1) Did the authors assess IL-10 production by DC10? Additionally, were type 1 regulatory CD4 T cells (Tr1) induced by DC10 in this experimental setting? Addressing these questions would provide further mechanistic insights.

We appreciate the suggestion of the reviewer. Unfortunately, the limitation of frozen PBMCs and fresh mucosal samples do not allow us to validate this hypothesis. This is an idea that was already formulated by our team, and we are planning future studies where we will address these concepts. Nevertheless, we are thrilled to see that the reviewer is supporting our ideas.

Minor comments:

Results (Lines 151, 159): The references to supplementary figures should be corrected: Supplementary Fig. 1a should be 2a, and Supplementary Fig. 1b-c should be 2b-c.

We have corrected the typo. Thank you.

Line 179: Please specify that the analysis was performed 2–3 weeks following the last vaccination.

line 179 is reporting the time-point of this analysis.

NATIONAL CANCER INSTITUTE
Center for Cancer Research

Response to Reviewers
NCOMMS-25-07875A

We thank the reviewers again for their input on the manuscript **HIV vaccine candidate mucosal immunity in female macaques augmented by Δ V1gp120 formulated in ALFQA adjuvant** by Bissa, *et al.* We are pleased that all previous comments have been satisfied.

REVIEWERS' COMMENTS

Reviewer #2 (Remarks to the Author):

The authors have addressed reviewer concerns to improve the quality and clarity of the work. The new statements in the paper should be reviewed and modified for improved style/grammar. For example, line 149 should read "the ALFQA and Alum groups..." and line 521 should read "Here, we identified the important role of..." etc.

We are pleased that the reviewer was satisfied by our previous revision of the paper. We have further reviewed the new statements to improve their style and grammar.

Reviewer #3 (Remarks to the Author):

The authors have thoroughly addressed all the points raised by this reviewer in the initial round. I am satisfied with the revisions and appreciate the effort invested in improving the manuscript. Thank you, and congratulations on your work.

We are pleased that the reviewer was satisfied by our previous revision of the paper.